# A universal all-solid synthesis for high throughput production of halide perovskite

Luyao Zheng[1], Amin Nozariasbmarz [1]✉, Yuchen Hou [1], Jungjin Yoon[1], Wenjie Li[1], Yu Zhang [1], Haodong Wu[1], Dong Yang [1], Tao Ye [1], Mohan Sanghadasa[2], Ke Wang[1], Bed Poudel [1]✉, Shashank Priya [1]✉ & Kai Wang [1]✉

Halide perovskites show ubiquitous presences in growing fields at both fundamental and applied levels. Discovery, investigation, and application of innovative perovskites are heavily dependent on the synthetic methodology in terms of time-/yield-/effort-/energy- efficiency. Conventional wet chemistry method provides the easiness for growing thin film samples, but represents as an inefficient way for bulk crystal synthesis. To overcome these, here we report a universal solid state-based route for synthesizing high-quality perovskites, by means of simultaneously applying both electric and mechanical stress fields during the synthesis, i.e., the electrical and mechanical field-assisted sintering technique. We employ various perovskite compositions and arbitrary geometric designs for demonstration in this report, and establish such synthetic route with uniqueness of ultrahigh yield, fast processing and solvent-free nature, along with bulk products of exceptional quality approaching to single crystals. We exemplify the applications of the as-synthesized perovskites in photodetection and thermoelectric as well as other potentials to open extra chapters for future technical development.

Halide perovskites and their versatile transforms have emerged as a new material class over multiple research fields prevailing from photovoltaics[1] to photodetection/photo-sensing[2], light emission/display[3], memristor/artificial synapsis[4], ferroelectric[5], photo-thermal conversion[6], thermoelectric[7], etc. The synthesis and manufacturing of perovskites are crucial to their structural and microscopic features at various scales, from lattice (e.g., distortion, strain, and defects), grain (e.g., size, orientation, and grain boundary), morphology and topography (e.g., homogeneity, surface roughness, and surface traps), all of which could synergistically affect the final physical property and their overall reliability in real implementations. For instance, different synthetic methodologies can lead to multiple-level disorder states and thereby a wide range of electronic attributes such as trap density ($10^9$ to $10^{19}\ cm^{-3}$) and carrier mobility ($10^{-6}$ to $10^2\ cm^2\ V^{-1}\ s^{-1}$) in the prototype methylammonium lead triiodide (MAPbI$_3$) perovskite[8] and

consequently a large discrepancy in device performance as well as the intrinsic stability of the material.

In physics, matter can exist in different states, including solid, liquid, gas, and plasma. The interstate transformation provides the opportunity of manufacturing a matter in an alternative but more convenient state for processing, followed by converting to the state of use. Prior manufacturing of halide perovskites mainly includes two scenarios: (i) liquifying the perovskites into solution followed by solidifying into film or crystals[1,9], and (ii) vaporizing the perovskites through high-vacuum thermal deposition techniques[10]. The former plays a dominant role in the field of thin film perovskite photo-electronics such as photovoltaic (PV) and light emission diode (LED). Albeit a decade's endeavor in innovation and development of solution-based processing with exciting deliverables, intrinsic methodological issues such as purity due to residual solvent molecules and impurity species from solution as well as the defects and lattice distortions that

[1]Materials Research Institute, The Pennsylvania State University, University Park, PA 16802, USA. [2]U.S. Army Combat Capabilities Development Command Aviation & Missile Center, Redstone Arsenal, AL 35898, USA. ✉e-mail: aln192@psu.edu; bup346@psu.edu; sup103@psu.edu; kaiwang@psu.edu

are 'frozen' during the nonequilibrium crystallization, remain to address. Further manufacturing difficulties including complex post-deposition treatment (e.g., anti-solvent, two/multi-step treatment, annealing, surface passivation), batch-to-batch consistency and reproducibility, up-scalability and mass-productivity, supply costs on solvent materials and energy input for solvent removing, as well as raw material waste from deposition (e.g., spin-coating), and eco-concern on solvent toxicity, can bring multiple risks to the technical transition. In parallel, vapor-based methods for perovskite synthesis have been attempted since the year 1997[11], with controllable purity, yield, and scalability as well as exemption of those solvent-related issues. Such a vapor technique could be more practically reliable, as learned from the prior success of commercialized organic light emitting diode (OLED), since such a 'dry' method can minimize moisture and oxygen concentration within the final products and, thereby, a higher level of efficiency and lifetime compared to the wet methods[12]. Unlike organics, vapor deposition of perovskites requires simultaneous sublimation of multiple precursors. While technically monitoring the evaporation rate of the low-molecular-weight ammonium halide (e.g., methylammonium iodide (MAI), formamidinium iodide (FAI)) and accurately controlling the stoichiometry remain challenging, which is also accompanied by other intrinsic issues such as low material usage (e.g., 10%) and unwanted decomposition through outgassing in high-vacuum due to the weak bonding nature of halide perovskites[13]. On the other hand, the current synthesis of bulk perovskite materials for applications in X-/γ-ray detection, photo-sensing, and thermoelectric still relies on the low-yield (10–35%) and time-consuming (0.1–10 mm$^3$ h$^{-1}$) wet chemistry methods[14]. Recent studies reveal the melting method to synthesize the bulk sample but critically limited to a specific composition (i.e., all-inorganic) due to the conflict between the high reaction temperature (over 500 °C) and the intermediate decomposition temperature (<250 °C) of general halide perovskites. All of these have already set up the barrier for not only mass production in the real application but also the high-throughput discovery of potentially more advanced perovskite materials.

Besides liquid and gas states, mechanically densifying the perovskites starting from the solid state with certain boundary effects, such as the plasma state, could provide a new synthetic route. Similar solid synthetic methods have been used in the oxide perovskite ceramics but mostly rely on an ultrahigh sintering temperature (e.g., over 1000 °C)[15] that can easily damage the halide perovskites. A traditional ceramic-sintering method, such as cold sintering, has been applied for synthesizing halide perovskite pellet[16–18]. However, these traditional methods lack sufficient thermal energy to activate crystal growth and generally lead to small crystalline grains with the presence of abundant voids and pin holes. Distinct from their oxide cousins, halide perovskites exhibit semiconducting (e.g., the intermediate bandgap of 1.6 eV of MAPbI$_3$) and soft lattice nature (e.g., Young's modulus of 10–20 GPa of MAPbI$_3$ vs 40–200 GPa of typical ceramics or metal alloys)[19]. The soft lattice nature allows sufficient deformation and densification of the perovskite powder precursors under mild pressure, and the semiconducting feature provides sufficient conductance of the current flowing internally through the whole green body to trigger other secondary or coupling effects microscopically that can be uniformly distributed within the sample. These material features provide convenience for both gentle electrical field and mechanical field to perform the synthetic densification of perovskite. For example, applying an internal electric field that can induce joule heat homogeneously distributed throughout the sample and simultaneously applying a uniform pressure to condensate the perovskite, could be a potential way to obtain high-quality perovskite crystals. Here, we demonstrate this first attempt at the universal synthesis of halide perovskites via an electrical and mechanical field-assisted sintering technique (EM-FAST) that directly densifies the perovskites from solid precursors into high-quality bulk crystals within minutes. This synthetic route for halide perovskites outpaces prior liquid- and vapor-based methods by nature of high throughput (synthesis rate of 0.5 cm$^3$ min$^{-1}$), 100% material usage, oxygen- and moisture-free, hazardous solvent-free, and capability of producing high-quality large bulk crystals in a short time.

## Results

### Proof of concept of FAST

The lab-customized EM-FAST (FAST, for simplicity) consists of a mechanical loading system with a high-power electrical circuit in a controlled atmosphere (Fig. 1a, "a. u." represents for "arbitrary unit"). Low voltage but high pulse current (1–10 kA) induces sufficient Joule heat and quickly elevates to a high temperature within minutes, where the current runs throughout the die and sample, causing secondary effects directly executed on the precursors. The unidirectional mechanical stress that is statically and uniformly applied to the plunger head further enhances these effects. And both stress and electric fields can be simultaneously applied on the sample during this non-equilibrium process. Notably, the halide perovskite is a well-matching material class because of its duality of soft lattice and semiconducting nature, allowing both stress and electric fields to actively affect the synthesis. Figure 1b shows the solid evolution of halide perovskites from powder precursors into the bulk crystal. The powder sample can be quickly heated up to a relatively low temperature (e.g., 200 °C for MAPbI$_3$) under high pressure for perovskites within 1 min and sintered for 2 min (Supplementary Fig. 1), followed by a cooling ramp (Supplementary Fig. 2). As a result, a bulk crystal with a diameter of 12.7 cm and thickness of ~0.2 cm is synthesized through FAST within 10 min of the whole process, which is significantly faster than the typical solution-based synthesis (<1 cm$^3$ day$^{-1}$)[14]. The densification can be understood by the mass transport mechanisms (Fig. 1c, d). The compressive pressure (*ca.* 50 MPa) leads to better contact between particles because of the small contact area (neck) and, thus, an enlarged localized pressure, triggering densification by grain boundary diffusion, lattice diffusion, and plastic deformation or grain boundary sliding[20]. Meanwhile, a pulse electric current directly runs through the sample, inducing internal heating concentrating at the neck (Fig. 1c). This is because halide perovskite can have a high carrier density (10$^{21}$ cm$^{-3}$)[21] at the crystal edge/grain surface, providing conductive channels for the pulse current to flow and induce the localized heat concentrating at the neck. This surface heating effect, plus the low ionic activation energy (e.g., 0.1–0.6 eV of MAPbI$_3$) in halide perovskites[22], triggers the mass transfer and grain merging between neighboring powders. Taking the classic perovskite MAPbI$_3$ as an example, we utilize the FAST to synthesize a highly densified bulk crystal. Starting from the precursors PbI$_2$ and MAI powders, we first employ a ball milling (BM) process to obtain the MAPbI$_3$ particles (Supplementary Figs. 3 and 4) and utilize the as-synthesized particles as the green body for FAST. Figure 1e, f compares initial powders and resultant samples, respectively. The as-milled particles show nonuniform dimensions from nm to µm scale as visualized by the scanning electron microscopy (SEM) (Fig. 1g), which then consolidate into uniformly sized grains (ca. 7 µm in Fig. 1h) by FAST. By tuning the synthesis conditions (e.g., stress and time), we have further obtained a highly compact crystal with a controllable grain size of up to 90 µm, where the crystallographic lattice features are further verified by high-resolution transmission electron microscopy (HR-TEM) (Supplementary Fig. 5).

### Performance evaluation of FAST-MAPbI$_3$

The X-ray powder diffraction (XRD) result reveals that the FAST-MAPbI$_3$ bulk crystal displays high crystallinity without impurity peaks of either precursor (Fig. 2a). In contrast to the MAPbI$_3$ powder (Supplementary Fig. 4c, d), FAST-MAPbI$_3$ also shows stronger (00*l*) orientation and the emerging of the *Pm$\bar{3}$m* cubic phase (as evidenced from

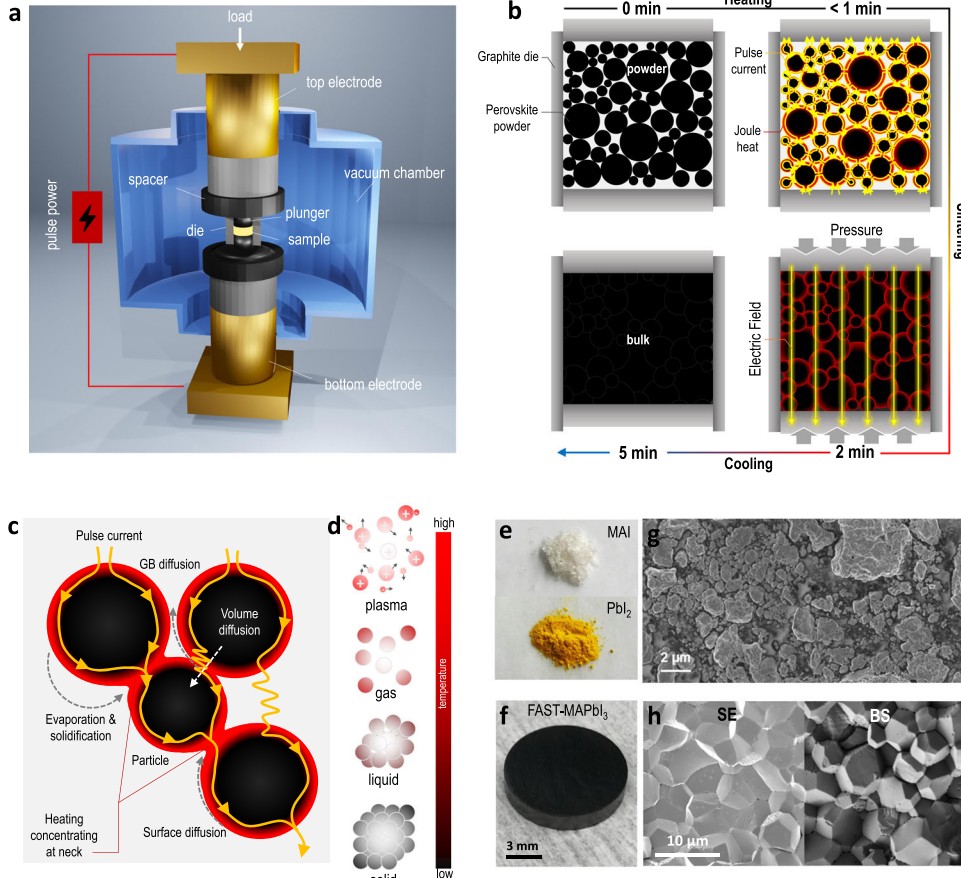

**Fig. 1 | Prototype FAST synthesis of MAPbI₃ perovskite. a** Setup configuration of FAST. In the chamber, a die with a loaded powder sample is sandwiched between the top and bottom electrodes, where the mechanical and electrical fields are applied. **b** Processing diagram showing the solid densification of halide perovskites in the graphite die. **c** Mass transport mechanisms involved in the sintering, including volume diffusion, evaporation and solidification, grain boundary (GB) diffusion, and surface diffusion. The red color surrounding the sphere surface indicates localized higher temperatures at the particle surface, which could activate the ionization of perovskites into plasma. **d** Matter states scheme showing the plasma state of the hot ionized gas of ions and electrons, which can hypothetically exist during the sintering of halide perovskites due to its low ionization energy threshold. Photos showing **e** the starting precursor powders of MAI and PbI₂, and **f** the FAST as-synthesized MAPbI₃ bulk disk. SEM images showing **g** the as-milled MAPbI₃ powders and **h** the cross-section of the FAST-synthesized MAPbI₃ bulk with secondary electron (SE) and backscattered (BS) detectors of the microscopy.

the presence of cubic phase scattering planes of (200) and (210) in Supplementary Fig. 6). This can be understood by the quick processing nature of EM-FAST which 'freezes' the high-temperature-favorable cubic phase and locks in this phase with surrounding tetragonal phases. The mechanical field further assists this phase lock-in effect, which may also induce certain lattice strain effects based on prior studies[23]. The highly symmetric cubic MAPbI₃ has a more ideal bandgap for PV and has been recognized as an efficient phase for solar cells[24]. Figure 2b compares the UV–vis absorption and steady-state photoluminescence (PL) spectra of FAST-, single crystal- and powder-MAPbI₃. Similar to the single crystal, the absorption edge of the FAST-MAPbI₃ reaches 852 nm, exceeding the edge of 780 nm of typical thin film counterparts, which can be ascribed to the in-direct band transfer[25] that leads to smaller optical bandgaps in thicker samples. It should be noted that the FAST sample shows a strong anti-Stokes shift suggesting higher energy states exist in the PL process. The presence of these higher energy states is more likely due to the cubic phase and lattice strain-induced Rashba splitting that enlarges the PL bandgap[26,27]. We calculate the optical bandgap to be 1.45 eV of FAST-MAPbI₃ from the corresponding Tauc plot (Supplementary Fig. 7), which is smaller than that of typical thin films (ca. 1.6 eV) but closer to that of single crystals (1.51 eV). The FAST-MAPbI₃ displays a sharp PL peak at 775 nm with a full width at half-maximum (FWHM) of 47 nm (Fig. 2b), within the range of solution-synthesized single crystals

(~20–70 nm)[14,28]. Compared to the powder nanocrystals, the FAST-MAPbI₃ also exhibits a PL blue-shift of 18 nm, indicating a larger quasi-Fermi level splitting between excited electrons and holes, i.e., reduced tail states near the band edge. This is also consistent with the time-resolved photoluminescence (TRPL) spectra displaying a longer intensity-weighted average lifetime (257 ns) of FAST-MAPbI₃ (Supplementary Note 1 and Supplementary Table 1). Notably, this value is over four-fold larger than its thin-film counterpart (61 ns) (Fig. 2c) and significantly larger than that of solution-synthesized single crystal (21 ns in the inset of Fig. 2c, and prior reports show a wide range from 109 ns[29] to <10 ns in a recent report[30]). We then estimate the trap density and charge carrier mobility of the FAST-MAPbI₃. Figure 2d shows the hole-only device configuration and the corresponding current–voltage (I–V) curve under dark conduction. Following the one-type carrier space-charge-limited current (SCLC) theory, we calculate the trap density of FAST-MAPbI₃ to be $5.4 \times 10^{10}$ cm$^{-3}$ with a hole mobility of 1.7 cm² V$^{-1}$ s$^{-1}$ (detailed in Supplementary Note 2). When comparing to the data set from prior literature (trap density-mobility plot in Fig. 2e), we find these values of FAST-MAPbI₃ located in a region closer to the single crystals rather than typical polycrystalline films. Particularly the trap density is over four orders of magnitudes lower than that of typical solution-processed MAPbI₃ thin films but close to that of single crystals. We ascribe these super properties to the highly compact crystals and the μm-scale grain size thanks to the FAST. In

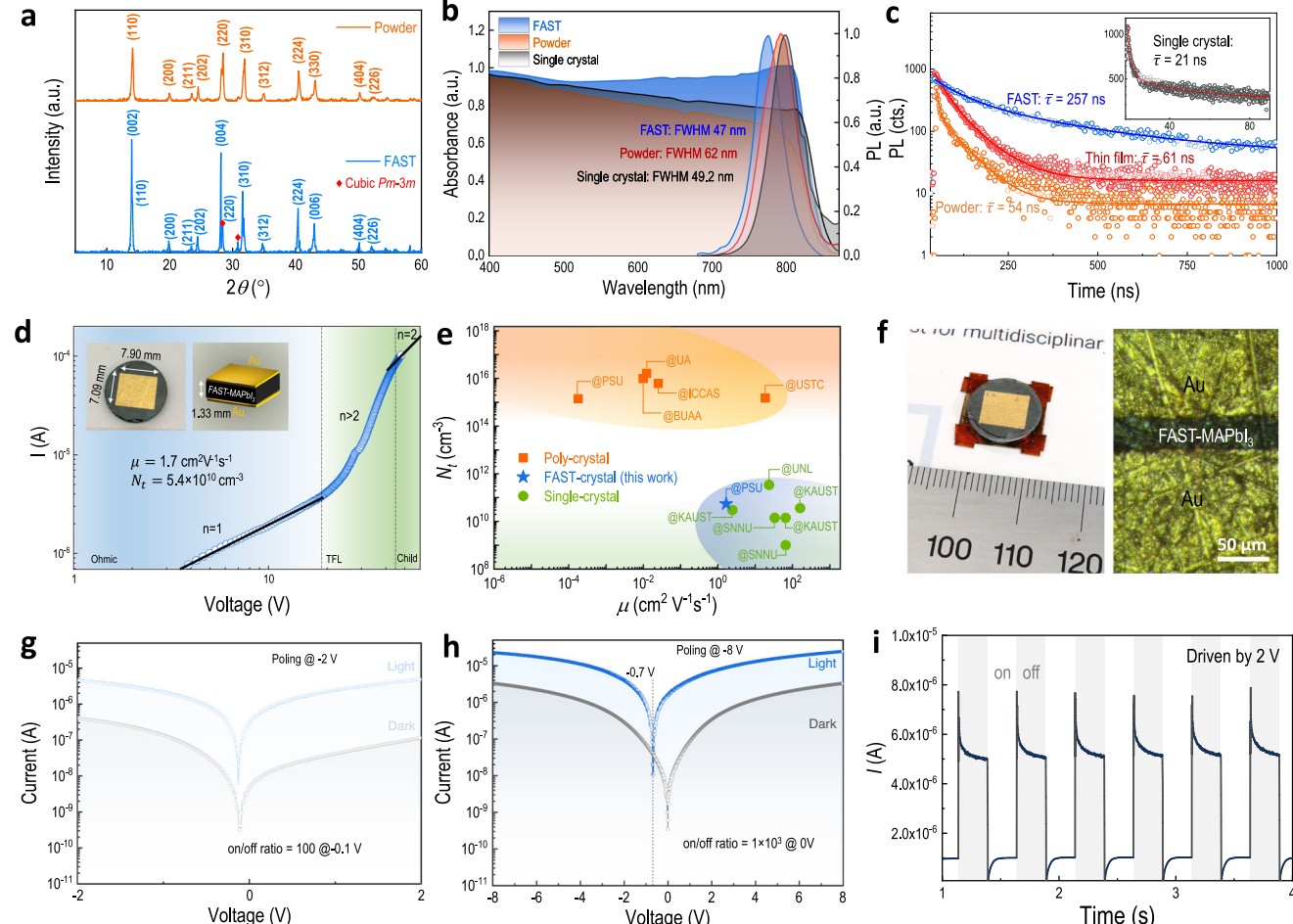

**Fig. 2 | Photoelectric evaluation of FAST-MAPbI₃. a** XRD spectra of as-milled MAPbI₃ powder and the FAST-synthesized MAPbI₃ bulk. The red diamond represents the peak from the cubic phase. **b** UV–vis absorption and steady-state PL spectra of FAST-MAPbI₃ and MAPbI₃ powder. **c** TRPL spectra of solution-processed MAPbI₃ thin film, as milled MAPbI₃ powder, and FAST-MAPbI₃. Inset: TRPL of MAPbI₃ single crystal. **d** I–V curve of the holy-only device under dark conditions. Inset: device configuration with marked dimension. **e** Comparison of trap density and mobility between FAST-MAPbI₃ and state-of-the-art solution-processed thin film and single crystal of MAPbI₃. The orange square, blue star, and green circle represent the poly-crystal, the FAST-crystal, and the single crystal, respectively. **f** Picture of the MIM transistor device and an optical microscopic image showing the channel length of 36 μm of the transistor device. I–V curve of the FAST photodetector under dark and light with poling bias of **g** −2 V and **h** −8 V, respectively. **i** Transient photocurrent of the FAST-MAPbI₃ photodetector under a periodic on/off light switching test.

general, the main difference between single crystal and polycrystal is the presence of grain boundary (GB) and lattice mismatch between neighboring grains. From the perspective of charge transfer, halide perovskites have been reported to have a polaron characteristic[31] that can help charge transfer across a long range of disordered states such as GB, which mitigates the detrimental role of GB on transport. In parallel, solution-processed single crystals have features such as the remaining solvent molecules within the crystal[32], the presence of surface trap[33], and point defects[34]. These features can jointly lead to additional trap states near the band edge, the disordering of the reciprocal lattice that can change the electronic band structure and, consequently, the electrical properties (e.g., charge carrier mobility, effective mass). These factors, from the downside of solution-grown single crystal and the upside of the FAST-polycrystal, can lead to the comparable performance in Figure 2e.

To further evaluate its photoelectrical behavior for device-level implementation, we build a lateral metal–insulator–metal (MIM, i.e., Au/FAST-MAPbI₃/Au) structured photodetector by a picosecond-pulse laser-scribing technique (Supplementary Fig. 8). Figure 2f shows a real device photo and an optical microscopic image of the transistor with a channel length of 36 μm. We measure the I–V curve in both dark and light conditions after a low-voltage poling process

(Fig. 2g, h) since the electric poling can trigger the formation of electric polarization of the device (by either ionic drifting, or dipole polarization, or both, discussed in Supplementary Fig. 9c, d) and thereby inducing an internal field to drive the photocarrier separation. By poling at −2 V, the FAST device displays an on/off ratio maximal of 100 at −0.1 V bias; the on/off ratio increases along with poling voltage (Supplementary Fig. 10), and by poling at −8 V, the device shows an open-circuit-voltage ($V_{OC}$) of 0.7 V and an on/off ratio of $10^3$ at 0 bias. The transient photocurrent of the FAST photodetector also displays a clear on/off switching performance (Fig. 2i) with a quick raising time of 745 μs (Supplementary Fig. 11), which is even faster than that of certain single-crystal-based photodetectors (Supplementary Table 2). We notice that at each light-switching point, there is a high current spike (Fig. 2i), which could be due to a resistor–capacitor (RC) effect that is typically observed in the transient response of OLED[35]. Overall, we utilize the poling method to break the electrical symmetry of the FAST MIM device, where such an asymmetry could also allow photocarriers of different types to drift towards opposite electrodes, leading to a photovoltaic behavior (Supplementary Fig. 9e). As a result, we measure the photocurrent density–voltage (J–V) curve of the FAST MIM device under Air Mass 1.5 sunlight. This lateral device exhibits a short-circuit-current ($J_{SC}$) of

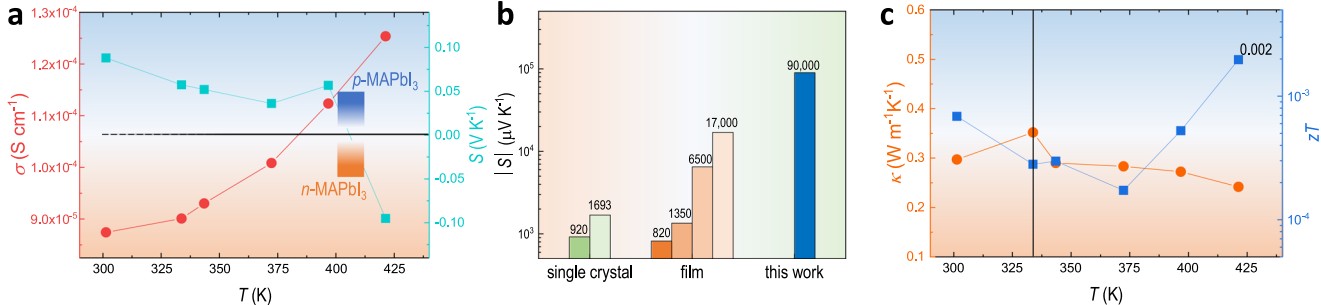

**Fig. 3 | Thermoelectric evaluation of FAST-MAPbI₃. a** Temperature-dependent electrical conductivity and Seebeck coefficient of FAST-MAPbI₃. **b** Comparison of the absolute Seebeck coefficient between FAST-MAPbI₃ and state-of-the-art single crystal and thin film of MAPbI₃, at room temperature. **c** Temperature-dependent thermal conductivity and $zT$ value of FAST-MAPbI₃.

6.3 mA cm⁻² and $V_{OC}$ of 0.71 V, with a power conversion efficiency (PCE) of 1.2%. Albeit of the relatively low PCE, the $J_{SC}$ and $V_{OC}$ values of the FAST MIM device still approach the values of the prior reported single-crystalline lateral device that has been optimized with surface trap passivation and charge selective buffer layers ($J_{SC}$ and $V_{OC}$ values of 0.78 V and 8.8 mA cm⁻², respectively)[36]. The relatively low PCE can be ascribed to the symmetric MIM configuration (low driving field from contact, although we apply the poling strategy to enhance the field), coarse gold contact (lacking proper buffer layers), and primitive interface. Nevertheless, despite of the polycrystalline nature, the FAST device exhibits excellent performance with figure-of-merits comparable to the device using single crystals.

Compared to thin-film perovskite, the FAST bulk sample with predesigned size and dimension can be a good platform for accurate quantification of bulk properties such as thermal conductivity ($\kappa$) and Seebeck coefficient ($S$). Figure 3a shows the temperature-dependent electrical conductivity ($\sigma$) and Seebeck coefficient ($S$) of FAST-MAPbI₃. The FAST-MAPbI₃ displays a relatively small electrical conductivity of $8.81 \times 10^{-5}$ S cm⁻¹ at 300 K (comparable to single crystal)[37] with a monotonously increased value to $1.26 \times 10^{-4}$ S cm⁻¹ at 421 K. The positive Seebeck coefficient at low temperatures reveals its p-type nature, consistent with most prior studies. Whereas the temperature increases to 421 K, there is a type conversion to n-type, which can be understood by the stoichiometric change due to the outgassing of MAI at high temperature and a consequent self-doping feature[38]. Notably, the FAST-MAPbI₃ shows a maximal Seebeck coefficient value of $9 \times 10^{4}$ μV K⁻¹ at room temperature, higher than the prior reported value in either MAPbI₃ thin film or single crystals (Fig. 3b) or typical thermoelectric materials (e.g., bismuth telluride alloy of ~200 μV K⁻¹ and nonmetal component of Selenium of ~900 μV K⁻¹) (Supplementary Table 3). Unlike typical thermoelectric materials with the metallic features of high carrier density and electrical conductivity, some halide perovskites have orders of magnitude lower carrier density that give rise to a higher Seebeck coefficient than typical degenerate semiconductors. This also leads to a small electronic contribution to thermal conductivity. As expected, the FAST-MAPbI₃ shows a low thermal conductivity of 0.3 W m⁻¹ s⁻¹ at 300 K (Fig. 3c), which is lower than that of typical solution-processed MAPbI₃ (Supplementary Table 4). The thermoelectric figure-of-merit, $zT$, value of FAST-MAPbI₃ is $2 \times 10^{-3}$ at 421 K (Fig. 3c). Compared to typical thermoelectric materials such as bismuth telluride, albeit that the MAPbI₃ halide perovskite shows more than 2-order-of-magnitude higher Seebeck coefficient and several times lower thermal conductivity, the significantly smaller (7-order-of-magnitude) electrical conductivity dominantly limits the $zT$ to a low value. Nevertheless, the FAST-MAPbI₃ sample shows great material quality with certain attributes (trap density, mobility, thermal conductivity, etc.) approaching or even exceeding those from single crystals (Fig. 2e and Supplementary Table 4).

## FAST compatibility to other perovskites

To explore the synthetic adaptability of the FAST to general halide perovskite materials, we synthesize other types of perovskites, including examples of alloy, lead-free, and two-dimensional (2D), and transparent perovskites, as well as their compatibility, to directly print bulk samples of pre-designed geometries and shapes. Figure 4a–p shows the crystal structural configurations and corresponding characterization results of various perovskite samples. To start with, we first observed that FAST makes it possible to alloy halide perovskites with solid materials (e.g., metal and carbon) that cannot be processed via the traditional solution method. Here we use the bismuth telluride as an example due to its great thermoelectric properties (high $zT$ of 1.2) as well as the difficulties to alloy it with halide perovskites using the conventional wet chemistry method. The resultant samples display two phases of Bi₂Te₃ and MAPbI₃ mixed sufficiently with highly compact crystal grain, as evidenced in the SEM images (Fig. 4b and Supplementary Fig. 12a). The energy dispersive X-ray spectroscopy (EDS) verifies the phase separation in the FAST-(Bi₂Te₃)₀.₉(MAPbI₃)₀.₁ ($x = 0.9$) with clear phase boundaries (Fig. 4c) and the perovskite phase is well dispersed within the Bi₂Te₃ matrix. We employ XRD spectroscopy for the identification of lattice planes in different samples. All the scattering peaks from the alloy sample can be assigned either to MAPbI₃ or Bi₂Te₃ (Fig. 4d), suggesting the alloy is a mesoscale mixture of individual MAPbI₃ or Bi₂Te₃ phases without creating a new phase. As expected, these FAST-(Bi₂Te₃)ₓ(MAPbI₃)₁₋ₓ ($x = 0.1$ and 0.9) alloys display intermediate thermoelectric properties within the range set by MAPbI₃ and Bi₂Te₃ (Supplementary Fig. 13). This is consistent to their physical blending nature and indicates no nontrivial coupling between the two materials. Although there are not exciting results from the alloy, this FAST method provides opportunities to alloy non-solution processible metals with perovskites for the first time. The heterojunction in the alloy can also be derived into a planar heterojunction via bi-layer interfacing between perovskite and other semiconducting materials of interest. An example of MAPbI₃/conductive polymer heterojunction is present in Supplementary Table 5. This could provide further opportunities for manufacturing the photoelectronic device directly on the basis of FAST-perovskite wafer/conductor heterojunction, following silicon wafer-based bottom-up manufacturing techniques (will be detailed later).

We also test the FAST synthesis for other compositions. Broader testing results and synthetic conditions are summarized in Supplementary Table 5, where at least two compositions in the same category are synthesized. For example, the lead-free perovskites MASnI₃ has emerged as a representative with optimal optoelectronic features such as the electronically delocalized states that enable higher mobility (10³ cm² V⁻¹ s⁻¹)[39] and smaller bandgap (1.24 eV, approaching to the ideal bandgap from Shockley–Queisser theory)[40]. The MASnI₃ belongs to the pseudo-cubic $Pm\bar{3}m$ space group, similar to the high-temperature phase of MAPbI₃ but substituting the B-site Pb²⁺ with

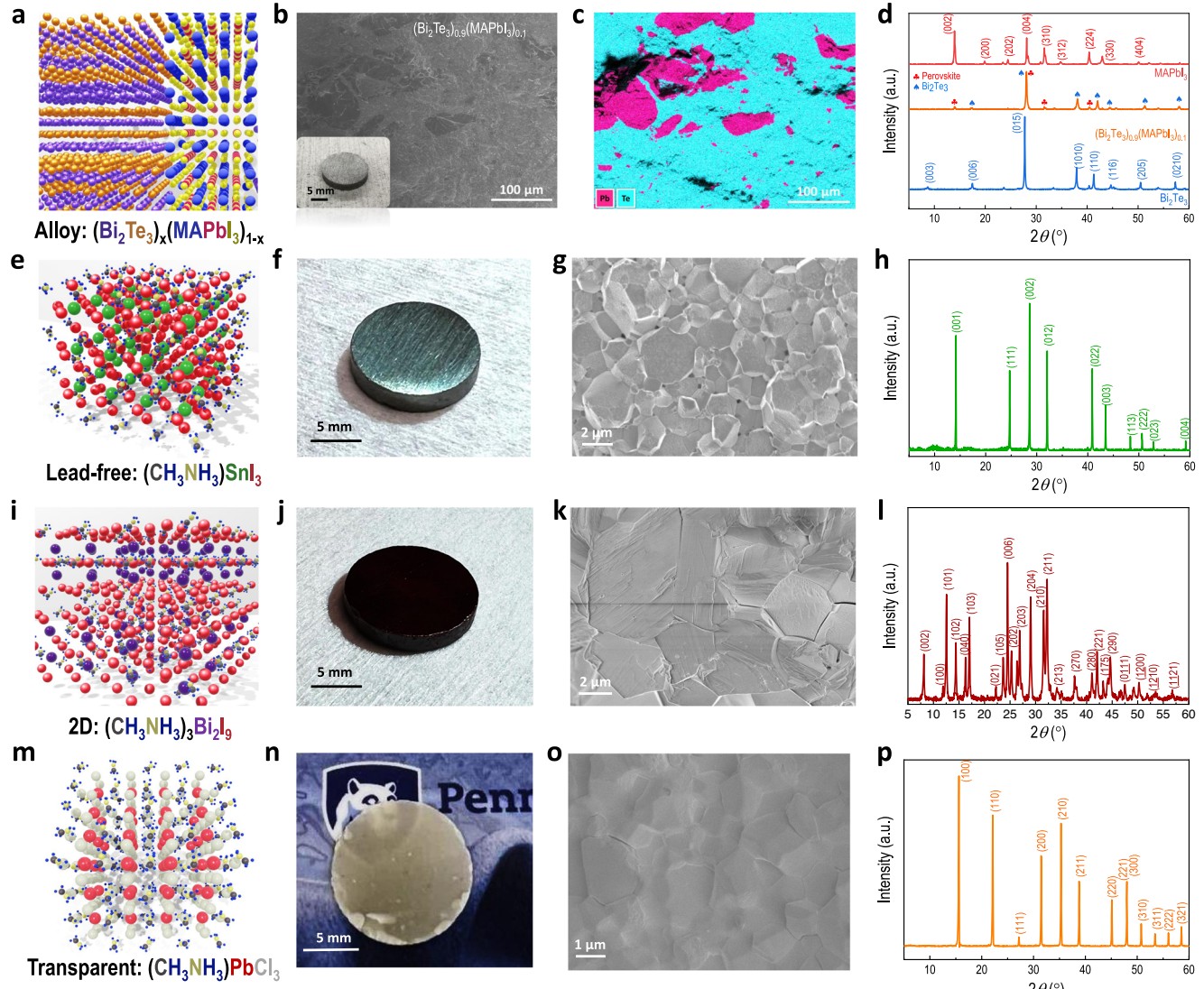

**Fig. 4 | FAST compatibility to broader types of halide perovskites.** Different compositions: $(Bi_2Te_3)_x(MAPbI_3)_{1-x}$ alloy: **a** scheme showing the configuration of the alloy lattice, where the atom is dressed in a color identical to that in the chemical formula. **b** Cross-sectional SEM image of the FAST-$(Bi_2Te_3)_{0.9}(MAPbI_3)_{0.1}$ with an inset of a photo of the FAST-$(Bi_2Te_3)_{0.9}(MAPbI_3)_{0.1}$-disk sample. **c** EDS images displaying the perovskite phase (pink) dispersed in the $Bi_2Te_3$ matrix (cyan), with a clear phase boundary. **d** XRD spectra of FAST-$MAPbI_3$ and FAST-$(Bi_2Te_3)_{0.9}(MAPbI_3)_{0.1}$ alloy, with comparison to that of $Bi_2Te_3$. The red club and blue spade represent the peak of $MAPbI_3$ and $Bi_2Te_3$, respectively. Lead-free perovskite of $MASnI_3$: **e** atomic scheme showing the pseudo-cubic structure, with atoms being dressed in a color identical to that in the chemical formula. **f** Photo of a real disk sample, **g** cross-sectional SEM image, and **h** XRD spectrum (with scattering peaks assigned to the corresponding lattice planes) of the FAST-$MASnI_3$. 2D perovskite of $MA_3Bi_2I_9$: **i** atomic scheme showing the 2D structure of $MA_3Bi_2I_9$, with atoms being dressed in a color identical to that in the chemical formula. **j** Photo of a real disk sample, **k** cross-sectional SEM image, and **l** XRD spectrum (with scattering peaks assigned to the corresponding lattice planes) of the FAST-$MA_3Bi_2I_9$. Transparent perovskite of $MAPbCl_3$: **m** atomic scheme showing the pseudo-cubic structure, with atoms being dressed in a color identical to that in the chemical formula. **n** Photo of a real disk sample showing a semitransparent feature, **o** cross-sectional SEM image, and **p** XRD spectrum (with scattering peaks assigned to the corresponding lattice planes) of the FAST-$MAPbCl_3$.

$Sn^{2+}$ in the $PbI_3$ octahedra (Fig. 4e). We synthesized the $MASnI_3$ from precursors of MAI and $SnI_2$ powder in a similar FAST-based route to $MAPbI_3$. Figure 4f displays the final disc sample where there is an obvious metallic shine (suggesting the existence of plenty of conduction electrons and thus induced plasmon frequency threshold effect in the sample)[41], which is consistent with the higher carrier density ($9 \times 10^{17}$ cm$^{-3}$) of this Sn-based material[42]. The SEM images of the FAST-$MASnI_3$ also show a compact packing of neighboring crystal grains (Fig. 4g) with an average grain size of ca. 2 μm. This crystalline feature is also supported by the XRD spectra (Fig. 4h), where the FAST-$MASnI_3$ displays high phase purity with a sharp scattering peak of small FWHM (0.29°) and all the peaks are well assigned to its pseudo-cubic $Pm\bar{3}m$ lattice planes[43]. It should be noted that prior wet-chemistry

methods reveal fast oxidation of Sn(II) to Sn(IV) in the air within a few hours and require a highly inert atmosphere for the whole synthesis process[44,45]. In comparison, the FAST automatically provides an enclosed atmosphere within the graphite die, and consequently, we obtain the high-purity disc sample with all manual operations in an ambient atmosphere.

To further test the FAST compatibility to more general halide perovskite compositions, we synthesize 2D perovskite of $MA_3Bi_2I_9$ (Fig. 4i–l and Supplementary Fig. 14) and its all-inorganic derivative of $Cs_3Bi_2I_9$ (Supplementary Fig. 15). These Bi-based 2D perovskites are lower-toxic and more stable materials than the lead family, and recently has been researched in fields of photovoltaics, photo- and X-ray detections with great performance[46]. Here, the FAST-$MA_3Bi_2I_9$

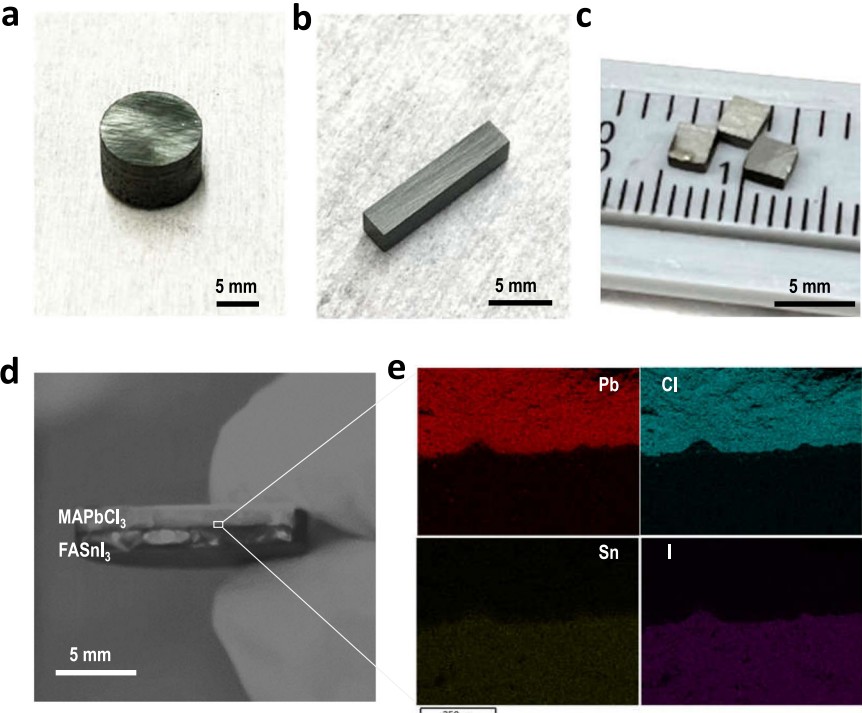

**Fig. 5 | FAST-based additive printed samples of different configurations.** FAST-MAPbI$_3$ samples as **a** 5 g column, **b** bar sample, and **c** cube samples. **d** Photo of bi-layer heterojunction of MAPbCl$_3$ and FASnI$_3$. **e** EDS mapping at the hetero-interface. Additional compositions can be found in Supplementary Fig. 17.

shows a dense feature (Fig. 4j) with a mass density of 4.10 mg cm$^{-3}$ close to the density functional theory (DFT) calculated value, which is consistent with its closely packed crystal grain as well as a micrometer scale grain dimension (Fig. 4k). The lattice structure of MA$_3$Bi$_2$I$_9$ is composed of face-sharing (BiI$_6$)$^{3-}$ octahedra layers with the MA$^+$ cations filling the space between these isolated (Bi$_2$I$_9$)$^{3-}$, and presenting an overall hybrid 2D layers consisting of alternating (BiI$_6$)$^{3-}$ and MA$^+$ ions (Fig. 4i). The XRD spectra verifies the phase purity with all the scattering peaks assigned to the corresponding lattice planes of the MA$_3$Bi$_2$I$_9$ in the hexagonal *P63/mmc* space group (Fig. 4l). Furthermore, we compare its experimental spectra with the calculated result in Supplementary Fig. 14, where scattering peaks of Omni (*khl*) planes are present, suggesting an isotropic nature of the FAST sample. Similar to MA$_3$Bi$_2$I$_9$, the FAST-Cs$_3$Bi$_2$I$_9$ also exhibit compact feature with an even larger grain size of 10 μm and great phase purity corresponding to the hexagonal *P63/mmc* space group (Supplementary Fig. 15a).

Similarly, the FAST can also synthesize transparent perovskites. As exemplified in Fig. 4m–p, we synthesized the MAPbCl$_3$ perovskite, which has a large bandgap of 2.9 eV and is thereby transparent to visible light. Figure 4n displays the semi-transparent sample where the translucency can be ascribed to the polycrystalline nature of the sample and an mm-thickness where the grain boundary can scatter the light and reduce the transmittance. This is evidenced by the μm scale grain size (Fig. 4o) and, consequently, a high density of grain boundaries. Nevertheless, the XRD shows high purity of the crystal (Fig. 4p). Overall, the FAST shows general compatibility with a wide range of halide perovskite materials. A quick microscopic and structural evaluation suggests a highly compact grain and high-level phase purity of all the FAST samples.

Besides the quick synthesis process and 100% material usage, the FAST also provides a bulk sample that can be geometrically customized through a predesigned die, which will allow future opportunities to manufacture a perovskite into certain shapes via pre-designed die/mold for direct assembly of a device (e.g., p–n thermoelectric unit in Supplementary Fig. 16d). One of such bulk examples is the thermoelectric module that consists of multiple thermoelectric legs. FAST-perovskite with customized designs can make it possible for mass production in a short time with negligible material waste (Supplementary Fig. 16). Figure 5a–c provides a few examples of such additive printed FAST-perovskites, where high-throughput production rate (Fig. 5a 2.5 g/min), arbitrary aspect-ratio (bar sample in Fig. 5b), and shape-designed molding (cube samples in Fig. 5c, i.e., route in Supplementary Fig. 16c) have been successfully attempted for a quick demonstration. In addition, we also constructed a heterojunction-structured perovskite by FAST. It should be noticed that there is a rare report on the ultrathick hetero-bilayer of halide perovskites[47]. As shown in Fig. 5d, a bilayer of MAPbCl$_3$/FASnI$_3$ disk has been obtained from a two-layer powder green body. The EDS elemental mapping (Fig. 5e) confirms the heterojunction of two different perovskites, and the SEM image (Supplementary Fig. 17) also reveals the intimate integration of these two perovskites. These results would provide the foundation for further exploration of the one-step bottom-up device fabrication that prints the MIM device or PN junctions, i.e., a ready device, directly from powder precursors.

## Exemplified application in thermoelectric

Rapid routes of obtaining novel materials with great crystallinity, purity, and morphology could save a huge time and efforts on synthesis, which could accelerate the material discovery for potential halide perovskites of new compositions and broader applications. Here, we take thermoelectric application as an example, starting from the theoretical calculation of a perovskite material of interest, followed by the rapid FAST, and discovery of superior thermoelectric properties, to demonstrate a closed-loop of research on thermoelectric materials. In contrast to typical metal alloys, halide perovskites have been recognized as a new type of thermoelectric material with low carrier concentration but ultrahigh Seebeck coefficient and ultralow thermal conductivity. Particularly, the Sn(II)-based perovskite exhibits higher conductivity among all the peers showing an overall characteristic closer to the phonon glass electron crystals (PGECs)[48]. Nevertheless,

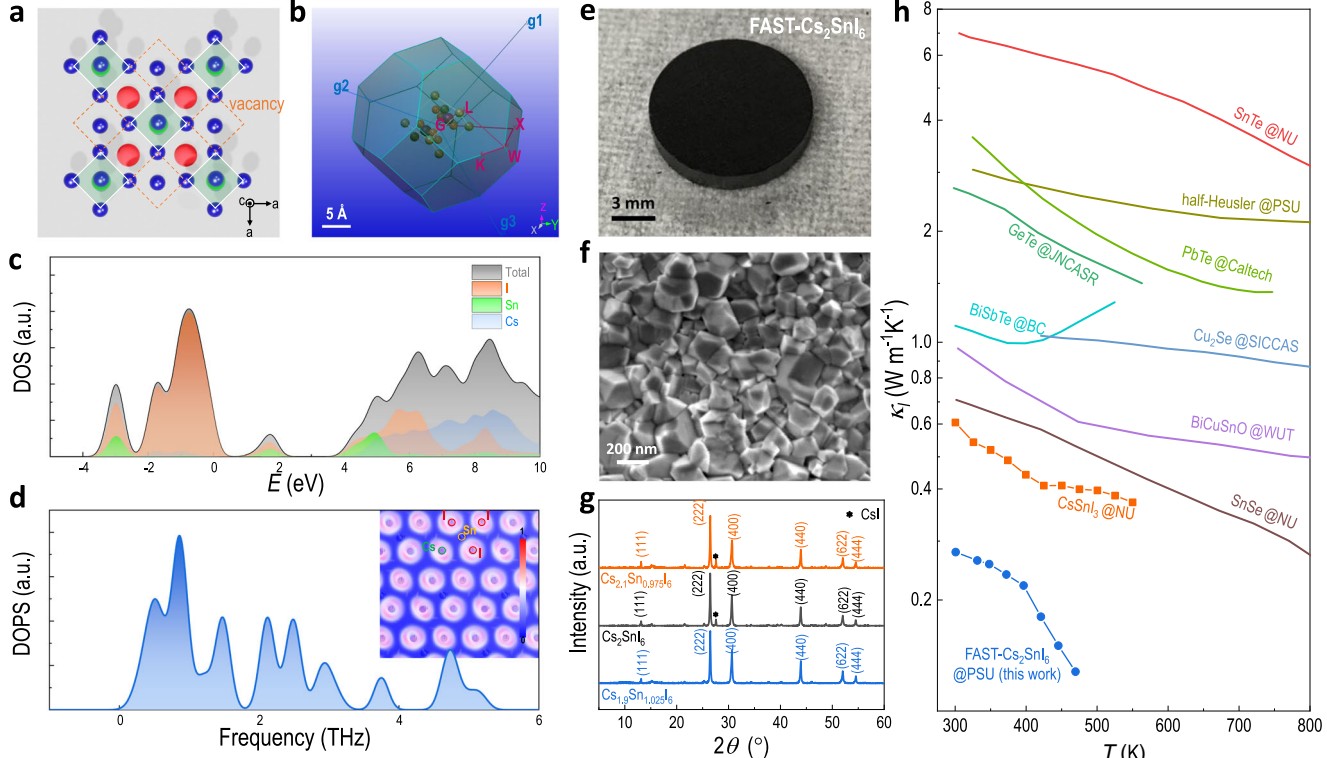

**Fig. 6 | Exemplified application of FAST-synthesized high-quality perovskites in ultralow thermal conductivity of intrinsic n-type FAST-Cs$_2$SnI$_6$.** **a** Lattice diagram of the vacancy-ordered feature. **b** The calculated Brillouin zone of the *fcc* lattice shows the symmetry points. **c** Total density of state (DOS). **d** Total density of phonon state (DOPS) of Cs$_2$SnI$_6$. **e** Photo of a FAST-Cs$_2$SnI$_6$ disk. **f** Corresponding cross-sectional SEM image. **g** XRD spectra of the FAST-synthesized Cs$_{2.1}$Sn$_{0.975}$I$_6$, Cs$_2$SnI$_6$, and Cs$_{1.9}$Sn$_{1.025}$I$_6$. The black asterisk represents the peak from CsI. **h** Temperature-dependent lattice thermal conductivity of the FAST-Cs$_{1.9}$Sn$_{1.025}$I$_6$ in comparison with other types of state-of-the-art thermoelectric materials.

prior studies on Sn(II)-based perovskite typically reveal a p-type feature and a quick oxidation to Sn(IV), which is a challenge for ambient application. Here we employ the FAST to synthesize a more stable Sn(IV)-based perovskite Cs$_2$SnI$_6$ with an intrinsic n-type characteristic simultaneously[49]. The Cs$_2$SnI$_6$ (space group $Fm\bar{3}m$) is a vacancy ordered double perovskites where the B-site vacancies are formed by the substitution of Sn(II) by Sn(IV) that leads to the sublattice regulation (Fig. 6a). DFT calculation (calculated Brillouin zone is present in Fig. 6b) reveals a direct band structure with a band gap of 1.25 eV (Supplementary Fig. 18b). Total density of state (DOS) indicates the valence band maximum (VBM) is mainly contributed by the I (5p orbital, Supplementary Fig. 18f), and the conduction band minimum (CBM) is mainly contributed by both I (5 s and 5p orbitals, Supplementary Fig. 18f) and Sn (5s orbitals, Supplementary Fig. 18d) states (Fig. 6c). The total density of phonon state (DOPS) (Fig. 6d) and phonon dispersion (Supplementary Fig. 19) reveals that there are no imaginary modes at negative frequency range and whereby indicates a stable nature of the material, distinguish it from its Sn(II) counterpart[7]. Similar to other halide perovskites[50], the total DOPS of Cs$_2$SnI$_6$ (Fig. 6d) also reveals no complete phonon band gap between optical and acoustic branches, indicating a chance of optical-to-acoustic conversion for the acoustic contribution to thermal conduction in typical halide perovskites. Nevertheless, the unique electron localization function (ELF) map (Fig. 6d and Supplementary Fig. 19c) shows a non-spherical electron density of the Cs and I atoms (due to the ionic nature of the Sn−I bond and structural asymmetricity because of the presence of the periodic vacancies in the lattice), which explains the origin of the large phonon anharmonicity and thereby a strong phonon scattering as well as a suppression on lattice thermal conductivity ($\kappa_l$)[51]. This is consistent with prior cases for other ordered crystals of ultralow

thermal conductivity[52,53]. Such a low $\kappa_l$, plus a low electrical contribution, could give rise to an ultralow thermal conductivity of Cs$_2$SnI$_6$. We then utilized the FAST to synthesize its bulk material, and as expected, we obtained a highly dense crystal (Fig. 6e and Supplementary Fig. 20) with closely packed grains (Fig. 6f). Interestingly, by using a stoichiometric ratio of precursor powder of CsI and SnI$_4$, we observed an impurity XRD peak of CsI at $2\theta$ of 27.58° (Fig. 6g). To avoid so, we adjust the ratio of precursors and found that addition of extra 2.5 mol% of SnI$_4$ could stabilize the crystal. A more detailed discussion has been incorporated in Supplementary Fig. 21. This is in accordance with the prior findings of the reaction between SnI$_4$ and moisture that can pull decomposition to proceed (Supplementary Note 3). Adding a sacrificial agent of SnI$_4$ can then maintain the phase purity of the resultant sample. This result also confirms that the FAST technique enables the development of non-stoichiometric compounds, which are not easy to obtain from other halide perovskites synthesis methods. We characterize the thermoelectric performance of the FAST-Cs$_2$SnI$_6$ perovskites by ZEM-3 series and laser flash analysis (LFA). Supplementary Fig. 22 presents the temperature-dependent electrical conductivity, Seebeck coefficient, thermal conductivity, and zT value of the FAST-Cs$_2$SnI$_6$-type perovskites using different precursor ratios. The FAST-Cs$_{1.9}$Sn$_{1.025}$I$_6$ with purer phase shows an inferior electrical conductivity of 0.05 S cm$^{-1}$ at 300 K, which is consistent with the cold-pressed Cs$_2$SnI$_6$ after annealing (ca. 0.01 S cm$^{-1}$)[54]. Along with the addition of extra CsI, the electrical conductivity of the final FAST sample increases, and the FAST-Cs$_{2.1}$Sn$_{0.975}$I$_6$ exhibits the highest electrical conductivity of 1.04 S cm$^{-1}$ at 469 K. All the FAST samples show a negative Seebeck coefficient revealing the n-type nature of the materials. It should be noted that using FAST, this n-type feature with high electrical conductivity is barely reported for halide perovskites[55]. Particularly for real

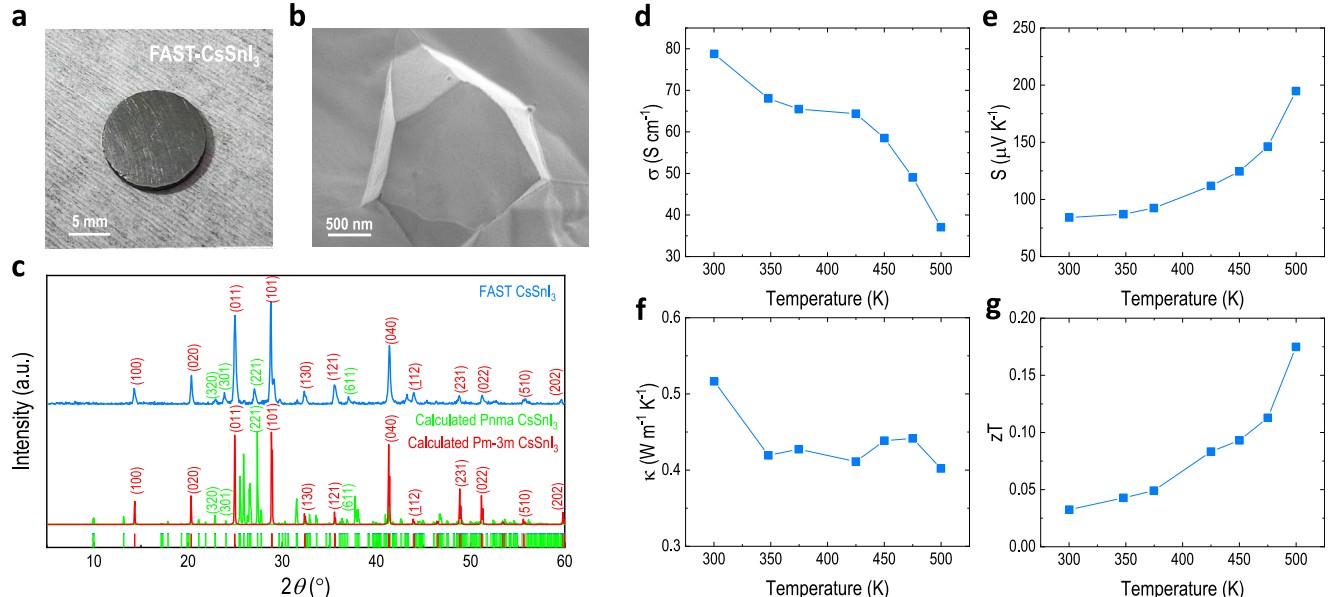

**Fig. 7 | FAST-CsSnI₃ with record zT among halide perovskites. a** Picture, **b** cross-sectional SEM, and **c** XRD spectra (with a comparison of calculated cubic and orthorhombic phases) of FAST-CsSnI₃ pellet. Temperature-dependent **d** electrical conductivity, **e** Seebeck coefficient, **f** total thermal conductivity, and **g** zT value of the FAST-CsSnI₃, respectively.

applications, it is critical to have p-type and n-type thermoelectric legs of congenial electrical and thermal conductivity to eliminate the joule loss and heat dissipation. Therefore, the FAST-Cs₂SnI₆-type perovskites would be a promising candidate to pair with those abundant p-type halide perovskites to reach a high performance for the thermoelectric device.

Notably, the FAST-Cs₂SnI₆-type perovskites exhibit ultralow thermal conductivities with the lowest value of 0.13 W m⁻¹ s⁻¹ at 469 K from the FAST-Cs₁.₉Sn₁.₀₂₅I₆ (Supplementary Fig. 23). Such a suppressed thermal transport implies the effective and unconventional phonon scattering in the system. The zT value of 0.055 attains from the FAST-Cs₂.₁Sn₀.₉₇₅I₆ at 469 K (highest value of intrinsic n-type perovskite, four-order-of-magnitude than solution-grown sample[56]). On this foundation, future endeavors, such as exotic doping, textural stretching, and alloying, can be applied to make a further boost in zT. In the meantime, the ultralow thermal conductivity of 0.13 W m⁻¹ s⁻¹ is obviously smaller than typical solution-processed 3D perovskites and even reaches the level of 2D multi-quantum-well (MQW) perovskites[57]. This can be understood by its small lattice and electronic thermal conductivity. We estimate the electronic thermal conductivity (via the Wiedemann–Franz relation) and lattice thermal conductivity (Supplementary Note 4). Briefly, due to the low electrical conductivity, the calculated electronic contribution to thermal conductivity is of 10⁻⁴ W m⁻¹ s⁻¹, indicating the overall ultralow thermal conductivity is dominantly contributed by the low lattice thermal conductivity. Figure 6h compares the lattice thermal conductivity ($\kappa_l$) of the FAST-Cs₂SnI₆ with other typical thermoelectric materials, displaying ultralow values ranging from 0.127 to 0.270 W m⁻¹ s⁻¹ from 470 to 300 K, 2-fold smaller than that of CsSnI₃ and significantly smaller than typical thermoelectric materials. This can be understood by the asymmetric electron localization function (ELF) and whereby phonon anharmonicity (Supplementary Fig. 19c) induced a stronger phonon scattering effect. Overall, the FAST enables the synthesis of more stable intrinsic n-type bulk samples, with properties superior to typical solution-processed samples.

Similarly, the theoretical calculation also predicts CsSnI₃, a promising thermoelectric material. In contrast to solution or vacuum growing processing, here we employ FAST to obtain this material. The sample shows a metallic shiny surface (Fig. 7a) with highly compact crystal grains (Fig. 7b). The CsSnI₃ normally has a cubic (Pm3̄m) phase at high temperature (500 K) and orthorhombic (Pnma) phase at room temperature (300 K). Interestingly as shown in Figure 7c, the FAST-CsSnI₃ exhibits both cubic and orthorhombic phases, with the major phase being the cubic. The cubic CsSnI₃ has a smaller calculated bandgap of 0.5 eV and a metallic-like feature. We investigated the thermoelectric behavior of the FAST-CsSnI₃ (Fig. 7d–g) and found a two-order-of-magnitude higher electrical conductivity than that of the abovementioned Cs₂SnI₆, but great Seebeck coefficient and thermal conductivity, which consequently contribute to a decent zT of 0.175 at 500 K (highest value compared to prior reports on solution-process halide perovskites[58]).

In summary, the rapid (electrical and mechanical) EM-FAST synthetic method can provide a new platform for making halide perovskites in a highly material- (100% yield), time- (0.5 cm³ min⁻¹), and effort- (solvent-free) efficient manner, with product crystals of high quality approaching or exceeding to those of single crystals. Additionally, there are several merits uniquely possessed by FAST-perovskite: (i) allowing the unsolvable dopant to be dispersed within the FAST-perovskite, which will bring future opportunities to incorporate/dope/blend metal, carbon, and other solution-unprocessable materials in with the perovskite; (ii) allowing the synthesis of functionally graded perovskite materials with a unidirectional compositional gradient that can be beyond the scope of the heterojunction of perovskite/conductive polymer in this work; (iii) allowing the direct additive printing of either multi-layer-structured optoelectronic device (e.g., solar cell, photodetector, and X-ray sensor) or leg-bridge-leg thermoelectric unit, as specified in Supplementary Fig. 24. We also carried out a stability test on our FAST-perovskite samples (Supplementary Figs. 25 and 26) which exhibit great stability that is better than typical solution-processed thin film samples. We anticipate this FAST-perovskite would open another dimension for high throughput material synthesis, future manufacturing directly printing devices from powder, and accelerating the material discovery of new perovskite compositions.

## Methods

### Materials

Methylammonium iodide (MAI, 99.999%), methylammonium chloride (MACl, 99.99%), and formamidinium iodide (FAI, 99.999%) were

purchased from Greatcell Energy Ltd. Lead (II) iodide ($PbI_2$, 99.999%), bismuth (Bi, 99.99%), tellurium (Te, 99.999%), antimony (Sb, 99.999%), bismuth (III) iodide ($BiI_3$, 99.998%), cesium iodide (CsI, 99.999%), and tin (IV) iodide ($SnI_4$, 99.999%) were purchased from Sigma-Aldrich. Tin (II) iodide ($SnI_2$, 99.999%) was purchased from Alfa Aesar. All the chemicals are used as received without any further purification.

## Synthesis

**Preparing of perovskites powders by ball milling.** Precursor materials were weighed according to stoichiometric ratios and transferred into a stainless-steel jar (bowl), with the whole process executed inside a glovebox with oxygen and moisture levels <0.5 ppm. High energy ball milling was conducted for 12–30 min accordingly by using a SPEX mixer/mill (Model 8000D, SPEX SamplePrep, Metuchen, NJ). The ground perovskite powders were collected and stored inside the glovebox for further characterization and/or consolidation.

**Electrical and mechanical field-assisted sintering technique (EM-FAST) for perovskite samples.** Perovskite powders were loaded into a graphite die (diameter of 12.7 mm). The loading process was executed inside the glovebox and then transferred outside for FAST. The whole FAST process for consolidation of powders was carried out by the spark plasma sintering machine (SPS, Dr. Sinter-625 V, Fuji, Japan) at a mild temperature (from 150 to 500 °C) under the pressure of ca. 50 MPa for 2–10 min, with an additional quick ramp of heating and cooling. The processing temperature varies upon different perovskite compositions, with an extra consideration of melting temperature as well as decomposition temperature of the material of interest. Briefly, the current in this report is along with the power surface and the neck between neighboring particles, i.e., the inner bulk sample (Fig. 1c, yellow curved arrows). Such an inner current can trigger joule heat localized at the particle surface. Upon merging of particles, the joule heat provides the energy to assist matter transfer across the boundary of the grains and helps crystal growth into larger sizes. It should be noted that although different perovskite compositions can have various range of electrical conductance, heat capacity/transfer properties, etc., which may require different current values to reach to certain temperatures. While the overall effect on the final sample quality is jointly determined by the thermal energy generated from the joule heat, the depth of melt or ionized region at the surface, time, and the equipment feature such as the type of die (steel or graphite), the size/dimension of die, and the amount of powder loaded (final thickness of disc), etc. This has been trialed in the case of the compositions covered by this work.

## Characterization

**Materials characterization.** Scanning electron microscopy (SEM) images were obtained by a field-emission SEM instrument (Zeiss Merlin LEO 1530). The energy dispersive X-ray spectrum (EDS) elemental mapping was measured with a SuperX EDS system under the SEM mode. The high-resolution TEM images were obtained by an FEI Titan3 G2 transmission electron scope (TEM) operating at an accelerating voltage of 80 kV. X-ray diffraction (XRD) data were collected on an X-ray diffractometer (Malvern Panalytical Empyrean) with Cu Kα radiation. UV-Vis absorption spectra were collected on a HITACHI UH4150 spectrometer. Photoluminescence (PL) spectra were acquired by using a fluorescence spectrometer (Edinburgh Instrument FLS 1000) at room temperature with a 506 nm excitation from a Xenon arc lamp. Time-resolved PL measurements were performed using a pico-second pulsed diode laser (505 nm excitation laser for $MAPbI_3$) as the excitation source and a time-correlated single photon counting (TCSPC) detector for signal collection. Space-charge-limited current (SCLC) measurement was performed with a Keithley 4200 source meter at room temperature in the dark, using a hole-only device configuration as specified in the manuscript. Electrical conductivity and Seebeck coefficient were simultaneously measured (Ulvac Riko

ZEM 3 System, Japan) using 2.0 mm × 2.0 mm × 12 mm bar sample. Thermal properties were determined by measuring thermal diffusivity with a laser flash system (LFA-467 HT HyperFlash, Germany). Specific heat was measured with a differential scanning calorimeter (NETZSCH DSC 214, Germany). Thermal conductivity, $\kappa$, was calculated from $\kappa = \alpha \rho C_p$, where $\alpha$, $\rho$, and $C_p$ are thermal diffusivity, density, and specific heat. The density was calculated by measuring the weight and thickness of the sample (with a fixed diameter of 12.7 mm).

**MIM device fabrication and characterization.** The MIM device for SCLC measurement utilized the configuration of gold/perovskite/gold. The gold electrode was deposited by the thermal vacuum evaporation method with an active area defined by a shadow mask. The gold layer thickness was controlled to 100 nm. The photo-transistor was manufactured by a pico-second laser scribing technique (350 nm, 20 W), where the laser scribed out the gold channel under carefully controlled duration, scan rate, and z-focus. The channel length is defined to be 36 μm after optimizing the laser scribing parameters. For solar cell demonstration on the MIM device, we utilized an asymmetric poling process by executing an 8 V bias on the transistor to trigger asymmetric polarization of the device. I–V curve in both dark and light conditions after a voltage poling process (−2 V, −5 V, and −8 V) was performed with a Keithley 4200 source meter. For the transient photocurrent of the FAST-$MAPbI_3$ photodetector, a function generator was used to generate a square periodic wave to modulate the on-off illumination of the LED light source. I-t curve was measured by a Tektronix MDO 3104 oscilloscope connected with a 50 Ω input impedance. J–V curve was obtained under Air Mass 1.5 sunlight for solar cell performance testing, where the light was provided by a standard 100 mW/cm² AM 1.5 G solar simulator (450 W Xenon lamp) which had been calibrated by a reference silicon cell covered by KG5 filter before each measurement.

**First-principle calculations.** The first-principle calculation of the electronic and phonon band structure was performed using CASTEP package[59,60]. Generalized gradient approximation (GGA) and Perdew-Burker-Ernzehof (PBE)[61] density functional theory (DFT) exchange-correlation potential was used with a cut-off energy of 500 eV for plane-wave basis set, a convergence threshold of $5 \times 10^{-6}$ eV per atom for energy and 0.01 eV Å$^{-1}$ for maximal force, respectively. Scalar relativistic approaches (Koelling–Harmon)[62] were used for relativistic effects treatment. For the calculation of electronic property, the $2 \times 2 \times 2$ k-point set within the gamma-centered Monkhorst–Pack scheme was employed to sample the Brillouin zone of the structures[63]. The phonon dispersion curves demonstrate how the phonon energy will depend on the q-vector along with the high symmetry directions within the Brillouin zone, which can be extracted from the neutron scattering technique for experimental. Here we also calculate the phonon density of state (DOS) and dispersion spectra from first principles.

**Recipe of the FAST-synthesized perovskites.** *$MAPbI_3$:* $MAPbI_3$ has a chemical formula of $CH_3NH_3PbI_3$. We obtained the $MAPbI_3$ powder from the BM process to stoichiometrically mix two precursors of MAI and $PbI_2$, and synthesize the sample by FAST at 52 MPa at 200, 250, and 290 °C, for 2 and 10 min, respectively. *The alloy of $(Bi_2Te_3)_x(MAPbI_3)_{1-x}$ (x = 0.1 or 0.9):* We firstly use BM to mix the powder of $MAPbI_3$ (obtained from the abovementioned BM methods from precursors of $CH_3NH_3I$ (99.999%, Greatcell Energy Ltd.) and $PbI_2$ (99.999%, Sigma-Aldrich) and $Bi_2Te_3$ (obtained according to prior reported methods[64]) following the stoichiometric ratio of the alloy. The mixture is then transferred to FAST to synthesize the disc, at a condition of 52 MPa at 200 °C for 2 min. *$MAPbI_3$/conductive polymers:* We use the as-synthesized FAST-$MAPbI_3$ disk (12.7 mm in diameter, ~2 mm in thickness) and conductive compression mounting compound (ProbeMet™)

to fabricate the heterojunction sample. Firstly, the ProbeMet™ powder is loaded into a die with a diameter of 30 mm, followed by a FAST-MAPbI$_3$ disk setting at the center. Then add the ProbeMet™ powder to fully cover the FAST-MAPbI$_3$ disc and slowly apply 0.5 MPa to compress the sample. After that, the heterojunction sample is obtained by FAST under 29 MPa at 150 °C for 2 min. As-synthesized heterojunction sample is further polished from one side until the FAST-MAPbI$_3$ disk is exposed at the center with the desired thickness. A buffer layer and transparent top electrode could be deposited on the top of the FAST-MAPbI$_3$ disk for PV applications, while the conductive polymer serves as the bottom electrode. *Bi-layer heterojunction*: The MAPbCl$_3$ and FASnI$_3$ powders are loaded into the die successively. Followed by FAST under 52 MPa at 200 °C for 2 min. *MASnI$_3$ and FASnI$_3$*: We firstly use initial powders of MAI or FAI (99.999%, Greatcell Energy Ltd.) and SnI$_2$ (99.999%, Alfa Aesar) to synthesize MASnI$_3$ or FASnI$_3$ powder by BM according to the stoichiometric ratio, then fabricate the disk by FAST under 52 MPa at 150 °C or 200 °C for 2 min. *(MA)$_3$Bi$_2$I$_9$*: We use the BM to prepare the (MA)$_3$Bi$_2$I$_9$ powder by mixing CH$_3$NH$_3$I (99.999%, Greatcell Energy Ltd.) and BiI$_3$ (99.998%, Sigma-Aldrich) with the molar ratio of 3:2. Then the 2D (MA)$_3$Bi$_2$I$_9$ disc is obtained by FAST under 52 MPa at 300 °C for 2 min. *MAPbCl$_3$*: Similar to FAST-MAPbI$_3$, the MAPbCl$_3$ powder is prepared by BM of MACl (99.99%, Greatcell Energy Ltd.) and PbI$_2$ (99.999%, Sigma-Aldrich). Followed by FAST under 52 MPa at 200 °C for 2 min. *Cs$_3$Bi$_2$I$_9$*: Similar to the FAST-(MA)$_3$Bi$_2$I$_9$, the Cs$_3$Bi$_2$I$_9$ powder is prepared by BM of CsI (99.999%, Sigma-Aldrich) and BiI$_3$ (99.998%, Sigma-Aldrich). Followed by FAST under 52 MPa at 500 °C for 2 min to synthesize the all-inorganic Cs$_3$Bi$_2$I$_9$ disk. *Cs$_2$SnI$_6$*: Firstly, the Cs$_2$SnI$_6$ powder is prepared by BM of CsI (99.999%, Sigma-Aldrich) and SnI$_4$ (99.999%) with the molar ratio of 2:1 or 2.1:0.975 (Cs-rich) or 1.9:1.025 (Sn-rich). Followed by FAST under 52 MPa at 300 °C for 2 min to synthesize the n-type vacancy-ordered Cs$_2$SnI$_6$ sample. *CsSnI$_3$*: The CsSnI$_3$ powder is prepared by BM of CsI (99.999%, Sigma-Aldrich) and SnI$_2$ (99.999%, Alfa Aesar). The FAST-CsSnI$_3$ sample is obtained by FAST under 52 MPa at 300 °C for 2 min. Other perovskites in this report use similar methods to prepare the powder and the FAST samples, where the temperature and pressure is controlled individually for each case.

## Disclaimer

*Full legal disclaimer*: This report was prepared as an account of work sponsored by an agency of the United States Government. Neither the United States Government nor any agency thereof, nor any of their employees, makes any warranty, express or implied, or assumes any legal liability or responsibility for the accuracy, completeness, or usefulness of any information, apparatus, product, or process disclosed, or represents that its use would not infringe privately owned rights. Reference herein to any specific commercial product, process, or service by trade name, trademark, manufacturer, or otherwise does not necessarily constitute or imply its endorsement, recommendation, or favoring by the United States Government or any agency thereof. The views and opinions of authors expressed herein do not necessarily state or reflect those of the United States Government or any agency thereof. *Abridged legal disclaimer*: The views expressed herein do not necessarily represent the views of the U.S. Department of Energy or the United States Government.

## Data availability

All data supporting this study are available within the article and the corresponding Supplementary Information file. Source data are provided in this paper.

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

## Acknowledgements

The authors acknowledge the support of the Material Characterization Lab (MCL) and the Nanofabrication Lab (NL), Materials Research Institute (MRI), Penn State. Kai.W. and Y.C. acknowledge Dr. Bangzhi Liu from NL, Penn State, for his contributions in SEM measurement and Dr. Congcong Wu from Hubei University for his contributions in MAPbI3-single crystal-related discussion. L.Z. acknowledges the support from NSF/IUCRC: Center for Energy Harvesting Materials and Systems (CEHMS) through award number IIP-1916707. This material is based upon work supported by the U.S. Department of Energy's Office of Energy Efficiency and Renewable Energy (EERE) under the Solar Energy Technologies Office Award Number DE-EE0009364 (H. Wu). Y.H. and Kai. W. acknowledge the financial support from the Air Force Office of Scientific Research (AFOSR Award Number FA9550-20-1-0157). S.P. acknowledges the support through National Science Foundation through award number DMR-1936432. A.N. acknowledges the support through the Office of Naval Research (ONR) through award number N00014-20-1-2602. B.P. acknowledges the support through the Army Research Office through award number W911NF1620010 (DARPA Matrix). J.Y. and T.Y. acknowledge the support through the ARMY RIF Program through award number W911W6-19-C-0083. D.Y. acknowledges the support through NSF CREST Center for Renewable Energy and Advanced Materials (CREAM).

## Author contributions

S.P. designed overall the concept of nonequilibrium manufacturing. Kai W., A.N., B.P., and M.S. specified the FAST idea and supervised the whole

project. L.Z. and A.N. conceived the idea of FAST synthesis of halide perovskite, and performed the experiment. L.Z., A.N., Y.H., D.Y., T.Y., and Kai W. performed the experimental characterization analysis and theoretical calculation. L.Z., Y.H., T.Y., J.Y., and H.W. designed and fabricated the photoelectronic device and performed the device characterizations. A.N., W.L., Y.Z., and B.P. performed the thermoelectric analysis. Ke W. performed TEM characterization. Kai W. and L.Z. prepared the first version of the paper. All the authors have discussed the research data, reviewed the paper, and provided comments.

## Competing interests

Patent disclosure is underfilling. The authors declare no competing interests.
