## [Peer Review File · Nature Communications]

A universal all-solid synthesis for high throughput production of halide perovskiteREVIEWER COMMENTS

Reviewer #1 (Remarks to the Author):

Overall, this is an interesting demonstration, and a rapid way of making polycrystalline pellets. However, a few points need clarification before being considered:

- Solid-state reaction routes have been widely investigated for perovskites. The authors should discuss the key innovations that differentiate their method to other pellet fabrication method (e.g., DOI: 10.1002/adma.202001981; <http://dx.doi.org/10.1038/nphoton.2017.94>). For instance, the electrical current effect is interesting here, what exactly does it do is ambiguous in the current MS. Could this vary depending on the conductivity, for instance MASnI_3 vs MAPbCl_3 ?

- More clarifications are needed for the phase-locking feature of MAPbI_3 by FAST method. The tetragonal peak features in Fig. 2a in the FAST sample still dominate the spectrum; and the crystal shapes shown in Fig. 1d still look like non-cubic structure.

- If the cubic phase is locked by the rapid high temperature processing, an internal strain may build up upon cooling since MAPbI_3 will transition to tetragonal phase at RT. Will this strain affect the overall stability?

- Fig. 2b plots the absorption and PL comparing FAST and powder samples. However, while the absorption edge of FAST red shift, its PL blue shifted above the gap. The current interpretation is too simple, and this PL peak position is far above the tail states below the gap. Does PL probe only the surface? Maybe inner part of the crystal should be probed as well (by cleaving). Does strain play a role here? This point needs some clarifications and perhaps more experimental support. The statement in line 138 "...which could be due to the transition between..." is quite confusing. Should be elaborated and clarified.

- Material stability should be investigated for the FAST grown pellets. Do these materials have better environmental stability? In particular, Sn-based perovskites are easily oxidized. The authors argued that the synthesis process is oxygen free, but what about post-synthesis stability?

Reviewer #2 (Remarks to the Author):

The authors developed a method for the all-solid-state synthesis of perovskite bulk crystals under high voltage and electric fields, which is called the field-assisted sintering technique (FAST). Different kinds of perovskite bulk crystals can be synthesized by this method. This work measures the properties of FAST-MAPbI₃ crystals (crystallinity, carrier mobility, trap density, thermoelectric properties, etc.), and exemplify the applications of FAST-perovskites in detection and thermoelectric. This FAST synthesis method provides a new platform for making halide perovskites, while some major revision is necessary before the manuscript is recommended for publishment in Nature Communications. Suggestions are as follows:

1. In line 135, page 5, FAST-MAPbI₃ bulk crystal can still maintain metastable state after cooling. Please provide relevant data and give a reasonable explanation.

2. In line 151, page 5, the average lifetime of FAST-MAPbI₃ is two times longer than that of solution-synthesized single crystals, relevant test data should be provided, and the cited references should be updated (a 2016 reference cited here is too old). Likewise, in Figure 2B, the authors should provide relevant test data for single crystals.

3. In line 155, page 5, the manuscript claims that the trap density and carrier mobility of FAST-MAPbI₃ are close to that of single crystals, but in fact the trap density is one order of magnitude lower than some single crystals, and the carrier mobility is two orders of magnitude lower (Fig. 2E). It is difficult to explain that the quality of FAST-MAPbI₃ crystal is comparable to single crystal. Please give a reasonable explanation.

4. In line 180, page 6, the FAST MIM device has a PCE of 1.2%, only 60% of the single-crystalline lateral device (2% PCE), and it's hard to say that the two values are comparable.

5. (1)The optical band gap of FAST-MAPbI₃ is smaller than that of single crystals (line 141, page 5).

(2)The average lifetime of FAST-MAPbI₃ is longer than that of single crystals (line 151, page 5).

(3)The Seebeck coefficient of FAST-MAPbI₃ is higher than that of thin films and single crystals (line 196, page 7).

Please give reasonable explanations for the above three points.

6. The manuscript uses a lot of space (page 11, line 333 to page 12, line 357) to describe how to regulate the ratio of CsI and SnI₄, which is not related to the topic of the article, and the author is advised to move it into Supplementary Materials.

Reviewer #3 (Remarks to the Author):

The manuscript by Zheng et al. reported the universal synthesis of halide perovskites via a field-assisted sintering technique that directly densifies the perovskites from solid precursors into high-quality bulk crystal within a few minutes. This method could be a powerful tool to prepare various halide perovskites. Although the manuscript showed many good results, some details, especially the mechanisms, should be further discussed and clarified. In this regard, a major revision is needed before being considered to get published in Nature Communications.

1. Why does the semiconducting and soft lattice nature of halide perovskites provide convenience for both gentle electrical field and mechanical field to perform the synthetic densification of perovskite simultaneously? The author should explain the mechanism clearly. Can this method be extended to other systems? Is it only suitable for materials with semiconductor and soft lattice properties?
2. As mentioned in the manuscript, "This surface heating effect, plus the low ionic activation energy (e.g., 0.1-0.6 eV of MAPbI₃) in halide perovskites, triggers the mass transfer and grain merging between neighboring powders", "The uniaxial mechanical stress further enhances these effects". According to these statements, the electric field is a unidirectional electric field and uniaxial mechanical stress, I wonder whether the perovskite is subjected to uniform stress in the whole system. If there is only uniaxial stress, the as-prepared perovskite should exhibit gradient density. Therefore, this approach might face some problems, such as whether the crystal quality and grain boundary of the upper and lower perovskites are different, which will seriously affect the properties and device performance.
3. The data presented in the manuscript indicate that the as-prepared perovskite is polycrystalline. There is no relevant mechanism explanation for why the performance of the as-prepared device based on such polycrystalline perovskites in figure 2 is comparable to that of the single crystal. In addition, the authors should provide some TEM data to prove the crystallization quality.
4. I do not quite agree with the relevant description about "heterojunction", which is more like a compound than a heterojunction. As shown in S17, at the interface of the MAPbCl₃ and FASnI₃, the density of the two materials is obviously different, and there are a lot of voids. The as-prepared material appears to show poor compactness. In addition, the synthesized heterojunction is too thick, can the perovskite heterojunction with few layers and periodic stacking be prepared?
5. The universality section should provide more data to demonstrate the advantages of this method. More than one perovskite of each type should be prepared, and the current data are not sufficient to prove that this is a general method.

6. The experimental content related to the stability of perovskite prepared by this method should be added to the manuscript.

7. The TRPL exhibits two decay paths, the slow and fast components should be fitted separately, and an explanation should be provided.

Reviewers' comments are highlighted in brown.

Our responses are in black.

The additional or revised sentences cited from the revised manuscript or Supplementary Information are highlighted in blue.

REVIEWER COMMENTS

Reviewer #1 (Remarks to the Author):

Overall, this is an interesting demonstration, and a rapid way of making polycrystalline pellets. However, a few points need clarification before being considered:

Our response: We greatly appreciate reviewer #1 for spending time reviewing this work. Thank you very much! Meanwhile, we are also very grateful for the valuable comments and suggestions that are quite helpful in improving our manuscript. We also feel excited that the reviewer finds that "this is an interesting demonstration, and a rapid way of making polycrystalline pellets". On the technical part, we have carefully addressed all the issues, please see the following details.

- Solid-state reaction routes have been widely investigated for perovskites. The authors should discuss the key innovations that differentiate their method to other pellet fabrication method (e.g., DOI: 10.1002/adma.202001981; <http://dx.doi.org/10.1038/nphoton.2017.94>). For instance, the electrical current effect is interesting here, what exactly does it do is ambiguous in the current MS. Could this vary depending on the conductivity, for instance MASnI_3 vs MAPbCl_3 ?

Our response: We appreciate the reviewer's comments as well as the valuable references. We greatly thank reviewer #1 for advising us to highlight our key innovations. We have added correlated contents in the revised manuscript. And in the text below, we briefly describe our novelty in comparison to other techniques, i.e., cold pressing and hot pressing.

Compared to cold pressing:

Cold pressing is a typical method to sinter bulk materials, where the densified pellet is obtained by applying a static pressure to the well-grinded micro-crystals with a certain assisted solvent, as mentioned in prior researches.¹⁻³ The principle is the mechanical compression that leads to densification. This could lead to some voids and pinholes in the resultant samples. In comparison to the FAST method in this work, those cold pressing methods do not contain extra fields such as electric field or the presence of plasmonic states during the process. Therefore, the development of crystalline grain growth remains difficult in the cold pressing method. As a result, there remain small grains, voids and pinholes in the final product. **Figs. R1** (adapted from prior reports) shows these features.

Fig. R1 SEM image of $\text{MA}_3\text{Bi}_2\text{I}_9$ **a** before and **b** after cold pressing¹. Comparing before and after pressing, the crystalline grain remains as similar size. **c** SEM cross-section of a sintered MAPbI_3 wafer. Inset: higher-magnification image. White circles indicate clear boundary and the existence of voids between the interconnected microcrystals³, adapted with permission of © 2020 WILEY-VCH Verlag GmbH & Co. KGaA, Weinheim; Copyright © 2017, Nature Publishing Group.

Compared to hot pressing:

Another typically used methods for solidification is a hot-pressing process, where heat is provided during the mechanical pressing. However, for halide perovskite, the decomposition temperature of certain composition (e.g., MASnI_3 and MAPbI_3) is lower than the melting point⁴. And the hot plates cover the outside

surface of the sample and thus the heat is from external sources. Heat transfer from outside to inside can lead to a gradient of temperature rendering inhomogeneous feature over the sample. To the best of our knowledge, directly using hot pressing to sintering perovskite pellet remains challenging and so far, we haven't seen related reports on this scenario.

Our method:

In contrast, we introduce both electrical and mechanical field in our sintering process, which initiates multiple mass transport mechanisms, e.g., volume diffusion, evaporation and solidification, grain boundary (GB) diffusion, and surface diffusion (**Fig. R2a**). This can lead to more densified crystal grains with larger size (**Figs. R2b-d**).

Fig. R2 a Mass transport mechanisms. **b** The cross-section SEM of the FAST-synthesized MAPbI_3 bulk with 200 °C and 2 mins. **c** The cross-section SEM of the FAST-synthesized MAPbI_3 bulk with 290 °C and 10 mins. **d** The cross-section SEM of the FAST-synthesized $\text{MA}_3\text{Bi}_2\text{I}_9$ bulk with 300 °C and 2 mins.

We have revised manuscript accordingly (on Page 3 and 4)

“...Traditional ceramic-sintering method such as cold sintering has been applied for synthesizing halide perovskite pellet^{16–18}. However, these traditional methods lack sufficient thermal energy to activate crystal growth and generally lead to small crystalline grains with presence of abundant voids and pin-holes. ... The soft lattice nature allows sufficient deformation and densification of the perovskite powder precursors under mild pressure and the semiconducting feature provides sufficient conductance of the current flowing internally through the whole green body to trigger other secondary or coupling effects microscopically that can be uniformly distributed within the sample. For example, applying internal electric field that can induce joule heat homogeneously distributed throughout the sample and simultaneously applying a uniform pressure to condensate the perovskite, could be a potential way to obtain high-quality perovskite crystals. Here, we demonstrate this first attempt on the universal synthesis of halide perovskites via such an electrical and mechanical field-assisted sintering technique (EM-FAST) that directly densifies the perovskites from solid precursors into high-quality bulk crystal within minutes...”

Electrical current effect

The current in this report is along with the power surface and the neck between neighboring particles, i.e., inner the bulk sample (**Fig R2a**, yellow curved arrows). Such an inner current can trigger joule heat localized at the particle surface. Upon merging of particles, the joule heat provides the energy to assist matter transfer across the boundary of the grains and helps crystal growth into larger sizes.

Conductivity dependence

In terms of different material systems, we agree with the reviewer that there could be difference in the ability of electrical conductance, heat capacity, heat transfer, etc., among different sample materials. Nevertheless, the overall goal of this internal current is to trigger a localized joule heat in order to merge the neck of neighboring particles. During the synthesis, we monitor the temperature in real-time. That is, the current can be different for different sample materials to reach a certain temperature value. For example, in our

experiment, in order to reach a temperature of 200 °C, MASnI_3 needs a lower current than that of MAPbCl_3 . Qualitatively, we find that the higher conductivity of perovskite, the lower applied current to initiate the joule heating to raise the temperature.

In parallel, it should be noticed that the temperature-current dependence during FAST also depends on the type of the die (steel or graphite), the size of die, and the amount of powder loaded (final thickness of disc), etc. The temperature is critical to determine the quality of resultant perovskites, with which we also considered the melting temperature and decomposition temperature for different type of perovskites (e.g. 200-290 °C for MAPbI_3 , 150-250 °C for MASnI_3 , 120-220 °C for MAPbCl_3). The crystal merging and growth rely more on the temperature (affecting the depth of melt or ionized region at surface), and time (influencing the degree of completion of annealing). Both of them are tuned to optimize the quality of resultant perovskites.

To make it clear, we have added the following explanation in the revised version (section of Methods):

“...Briefly, the current in this report is along with the power surface and the neck between neighboring particles, i.e., inner the bulk sample (**Fig. 1c**, yellow curved arrows). Such an inner current can trigger joule heat localized at the particle surface. Upon merging of particles, the joule heat provides the energy to assist matter transfer across the boundary of the grains and helps crystal growth into larger sizes. It should be noted that although different perovskite compositions can have various range of electrical conductance, heat capacity/transfer properties, etc., which may require different current values to reach to certain temperatures. While the overall effect on the final sample quality is jointly determined by the thermal energy generated from the joule heat, the depth of melt or ionized region at surface, time, and the equipment feature such as the type of die (steel or graphite), the size/dimension of die, and the amount of powder loaded (final thickness of disc), etc. This has been trialed in the case of the compositions covered by this work...”

- More clarifications are needed for the phase-locking feature of MAPbI_3 by FAST method. The tetragonal peak features in Fig. 2a in the FAST sample still dominate the spectrum; and the crystal shapes shown in Fig. 1d still look like non-cubic structure.

Our response: We thank the reviewer for pointing out this question. We do observe the co-existence of cubic and tetragonal phase in the final MAPbI_3 sample, although the ratio of cubic phase is very small. We understand the cubic phase at room temperature is metastable and it would have a trend in typical to revert back to the tetragonal. The locking may be ascribed to the intensified fields during the sintering, which will be discussed later in this response. Overall, we have added several descriptions in the revised manuscript.

On page 5, we have added the following description:

“...(as evidenced from the presence of cubic phase scattering planes of (200) and (210) in **Supplementary Fig. 6**). This can be understood by the quick processing nature of EM-FAST which ‘freezes’ the high-temperature-favorable cubic phase and locks in this phase with surrounding tetragonal phases. The mechanical field further assists this phase lock-in effect, which may also induce certain lattice strain effect based on prior studies²³.”

Co-existence of tetragonal and cubic phase

Generation of cubic during sintering: MAPbI_3 has a phase transition temperature of 55 °C from tetragonal to cubic. The generation of cubic phase can be understood by the high internal temperature caused by the joule heating during the synthesis as well as the extra mechanical stress. Prior researches also show that for pelletized MAPbI_3 sample (cold pressing), it would take more than 10 hours of annealing

at 250 °C to totally convert the tetragonal phase into the cubic⁵. In our study, with the temperature ramping up to 200 °C, the cubic phase of MAPbI₃ is formed at this high temperature. While this process is very quick (2 min), it would be rational that the dominant phase is still tetragonal.

Remaining of cubic after sintering: After cooling down, this cubic phase is still observable from XRD. **Figs. R3c and R3d** show the cubic phase with peaks of (200) and (210) presented at 2Theta of 28.19°, and 31.62°. Quantitatively, albeit there is co-existence of both cubic and tetragonal phases, the ratio of cubic is small. This may be due to the short sintering time that provides insufficient time to develop cubic phase, or the spontaneous recovering back to tetragonal phase (since cubic is thermodynamically unstable at room temperature).

In conclusion, tetragonal phase still dominates the phase, consistent to the XRD results and SEM in **Fig. 1d**. The interesting part could be the remaining cubic phase detected at room temperature. From our prior experience in solution processed film of MAPbI₃, after cooling the sample rarely shows any cubic phase at room temperature⁶. This difference leads us to reconsider if this cubic phase was locked in the pellet sample (as mentioned by reviewer #1) by some unknown reason (could hypothetically be the strain, bulk effect, or other unknown mechanisms, will be discussed in the answer to the next question).

Fig. R3 **a** XRD spectra of as-milled MAPbI₃ powder and the FAST-synthesized MAPbI₃ bulk. **b** Enlarged XRD spectra distinguishing the (110) and (002) scattering planes for powder and FAST sample. The FAST samples display the emergence of (00l) preferred orientation, compared to the powder sample. **c** and **d** Enlarged XRD spectra showing the emerging cubic phase in the FAST sample, as marked by the red diamond in the figure.

- If the cubic phase is locked by the rapid high temperature processing, an internal strain may build up upon cooling since MAPbI will transition to tetragonal phase at RT. Will this strain affect the overall stability?

Our response: Thank you for this inspiring question. We do agree with the reviewer #1 that an internal strain could be built-up not only due to the quick cooling phase lock-in effect, but also due to the locally intensified stress by the high pressure in the process. We have discussed the existence of cubic phase in the answer to the last question, from two aspects of (i) origin (generation of cubic) and (ii) remaining (remaining cubic in the sample at RT).

Strain role: We do appreciate reviewer guides us to think about the strain role in regarding with the remains of cubic in the FAST sample after cooling down to room temperature. Considering the difference in lattice parameter between cubic and tetragonal, the trend of cubic to proceed a phase transition from cubic to tetragonal at room temperature, as well as the counter that the surrounding tetragonal phases may inhibit such a cubic-to-tetragonal transition due to a densified bulk effect (i.e., the stable surrounding tetragonal grains confine the inner cubic phase in a limited space so that preventing it to change its structure/volume), the strain may remain within the sample.

Strain effect on stability: We have learned from literature that the compressive strain could increase the activation energy of ion migration^{7,8}, hence may enhance the sample stability. To understand this, supplementary data is added into the work. **Fig. R4** shows the stability result. Even after storing in the ambient for 2 months, the XRD results still exhibit identical spectrum to that of the fresh sample (no degraded impurities such as PbI_2 is detected). By enlarging the results, we observed that the original cubic phase (**Fig. R4b**) gradually converts into the tetragonal phase. This can be further visualized in **Fig. R4c**, that the (200) cubic peak in the fresh sample has an identical peak intensity to the neighboring (220) tetragonal peak. While after 2 months, the (220) tetragonal peak shows relatively stronger intensity than the (200) cubic peak.

Fig. R4 XRD results of MAPbI_3 pellet stored in the ambient atmosphere of 0h (fresh), 1h, 12h and 2 months. **a** XRD results of a EM-FAST MAPbI_3 pellet stored in the ambient atmosphere for different time (RH 35-80%, depending on daily whether at State College, PA, United State, temperature of ca. 25 °C controlled by lab). **b** Local magnification of the XRD results, showing the phase evolution from cubic (200) planes towards tetragonal (004) and (220) planes. **c** Comparison of fresh and 2-month age XRD, displaying the phase evolution.

We have added these results in the revised **Supplementary Information** (i.e., **Supplementary Fig. 25**). Detailed below:

“

Supplementary Fig. 25 Stability study of the FAST MAPbI_3 samples. **a** XRD results of a EM-FAST MAPbI_3 pellet stored in the ambient atmosphere for different time (RH 35-80%, depending on daily whether at State College, PA, United State, temperature of ca. 25 °C controlled by lab). **b** Local magnification of the XRD results, showing the phase evolution from cubic (200) planes

towards tetragonal (004) and (220) planes. c Comparison of fresh and 2-month age XRD, displaying the phase evolution. These results suggest robust feature of the EM-FAST MAPbI₃ samples (no obvious degradation even after ambient storage for 2 months). The origin may come from the hypothetical lattice strain in these samples, which could increase the activation energy of ion migration^{32,33}.

”

- Fig. 2b plots the absorption and PL comparing FAST and powder samples. However, while the absorption edge of FAST red shift, its PL blue shifted above the gap. The current interpretation is too simple, and this PL peak position is far above the tail states below the gap. Does PL probe only the surface? Maybe inner part of the crystal should be probed as well (by cleaving). Does strain play a role here? This point needs some clarifications and perhaps more experimental support. The statement in line 138 “...which could be due to the transition between...” is quite confusing. Should be elaborated and clarified.

Our response: Thank you for this inspiring question. Accordingly, we added an experiment and cleaved the sample to reveal inner surface for the PL. In the meantime, we also modulate the intensity of the excitation light to make it penetrate into different depths of the sample. However, the results in **Fig. R5** shows that the PL peak still sits around 774 nm, below the absorption edge of 852 nm. We also tested the PL for single-crystal sample (revised **Fig. 2b**), the absorption edge is similar while the PL peak is less blue-shifted compared to the FAST sample.

More-Red absorption edge: It should be noted that the absorption edge of >850 nm is more-red than typical thin film MAPbI₃ (780 nm). Prior reports have shown this could be due to the in-direct band transfer⁹ that leads to smaller optical bandgap. This theory is consistent to the fact that only in thicker samples, such a more-red absorption edge is observed.

More-Blue PL peak: To understand the strong anti-Stokes shift of PL, we think this result is directly related to the higher energy states in the sample. These higher energy states may be explained by the presence of cubic phase and the strain-related effect. In the MAPbI₃, the presence of heavy elements such as lead and iodine introduces a strong spin-orbit coupling (SOC) in the electronic structure. When inversion symmetry is absent (e.g., the case in the bulk of non-centrosymmetric boundaries and/or under strained conditions), the electrons feel an effective magnetic field due to the SOC. This interaction, known as the Rashba effect¹⁰, removes the electron spin degeneracy and splits the valence and conduction band edges, and eventually leads to some hypothetically higher energy states. As can be seen in **Fig. R5c**, the cubic phase contributes Rashba splitting with higher bandgap exceeding 1.65 eV (751 nm).

Jointly considering aforementioned effects, this strong anti-Stokes shift in the FAST-sample can be explained. Although beyond the scope of this work, we think further investigation is also needed here. While this FAST sample could be a good material platform for these strain-related studies.

Fig. R5 Steady-state PL of the MAPbI₃ sample with (a) cleaving process (each cleaving is about 20 μm) and (b) with different excitation light intensity (assuming higher intensity light can penetrate deeper of the sample according to Lambert Beer law). c Effective masses of electrons and holes in the cubic (red) and tetragonal (blue) MAPbI₃ phases versus the band gap, according to DFT results from prior report¹⁰. Adapted with permission from Springer Nature 2018.

Additional explanations have been added in the main text (page 5):

“...Similar to the single crystal, the absorption edge of the FAST-MAPbI₃ reaches 852 nm, exceeding the edge of 780 nm of typical thin film counterparts, which can be ascribed to the indirect band transfer²⁵ that leads to smaller optical bandgaps in thicker samples. It should be noted that the FAST sample shows a strong anti-Stokes shift suggesting higher energy states existing in the PL process. The presence of these higher energy states are more likely due to the cubic phase and lattice strain induced Rashba splitting that enlarges the PL bandgap^{26,27}. ...”

- Material stability should be investigated for the FAST grown pellets. Do these materials have better environmental stability? In particular, Sn-based perovskites are easily oxidized. The authors argued that the synthesis process is oxygen free, but what about post-synthesis stability?

Our response: Thanks for the important questions on the stability of the samples. We have incorporated the stability data in our **Supplementary Information** and added the discussions in the main text.

For the MAPbI₃ system, data is included in **Supplementary Fig. 25** and has been discussed in above text.

For the Sn-based system, we have added the following data and discussion in **Supplementary Fig. 26**.

Supplementary Fig. 26 Stability study of the FAST FASnI₃ samples. **a** XRD results of a EM-FAST FASnI₃ pellet stored in the ambient atmosphere for different time (RH 35-80%, depending on daily whether at State College, PA, United State, temperature of ca. 25 °C controlled by the lab). **b** Photography of the FASnI₃ sample with different ages from 0 h to 24 h, exposed in ambient air. A degradation chemical reaction set is inserted to show the compositional evolution.

First of all, it is shown that in the case of storage inside of a glovebox, even after 1 year, the sample does not show any impurities or degradation peaks in the XRD. While storing in the ambient air, after 12 h there is emergence of SnI₄ phase, and after 48 h there is emergence of SnO₂ phase. This evolution is consistent to the sequential degradation process in the equation set in **Supplementary Fig. 26b**. While from the photo, it is hard to observe obvious color change of the sample. This observation, together with the remaining scattering peaks of FASnI₃ in the 48-h age sample in **Supplementary Fig. 26a**, suggest there could be a protection layer formed by the degraded covering the surface, which can prevent further degradation of the inner sample.

Reviewer #2 (Remarks to the Author):

The authors developed a method for the all-solid-state synthesis of perovskite bulk crystals under high voltage and electric fields, which is called the field-assisted sintering technique (FAST). Different kinds of perovskite bulk crystals can be synthesized by this method. This work measures the properties of FAST-MAPbI₃ crystals (crystallinity, carrier mobility, trap density, thermoelectric properties, etc.), and exemplify the applications of FAST-perovskites in detection and thermoelectric. This FAST synthesis method provides a new platform for making halide perovskites, while some major revision is necessary before the manuscript is recommended for publication in Nature Communications. Suggestions are as follows:

Our response: We greatly appreciate reviewer #2 for reviewing, and providing many constructive comments to our work. We are also happy to see reviewer #2 finds this work as “a new platform for making halide perovskites”. Here we have carefully addressed all the issues in the original version. Please see the following answers in detail.

1. In line 135, page 5, FAST-MAPbI₃ bulk crystal can still maintain metastable state after cooling. Please provide relevant data and give a reasonable explanation.

Our response: We thank the reviewer for this comment. In principle, the MAPbI₃ has a phase transition temperature of 55 °C from tetragonal to cubic. At room temperature, the higher temperature favorable cubic phase will be metastable, and have a trend to convert into tetragonal phase. We have observed the existence of cubic phase in the FAST sample at room temperature (**Figs. R3c&3d**, shown in this response letter (answers to reviewer#1)).

“Fig. R3 c and d Enlarged XRD spectra showing the emerging cubic phase in the FAST sample, as marked by the red diamond in the figure.”

From the experimental results, these metastable cubic phases still remain in the sample after cooling. The hypothetical reason could be the phase locking effect that the FAST-generated cubic phase has been confined within the surrounding tetragonal medium even at room temperature (i.e., the stable neighboring tetragonal grains confine the inner cubic phase in a limited space so that preventing it to change its structure/volume).

On page 5, we have added the following description:

“...(as evidenced from the presence of cubic phase scattering planes of (200) and (210) in **Supplementary Fig. 6**). This can be understood by the quick processing nature of EM-FAST which ‘freezes’ the high-temperature-favorable cubic phase and locks in this phase with surrounding tetragonal phases. The mechanical field further assists this phase lock-in effect, which may also induce certain lattice strain effect based on prior studies²³.”

2. In line 151, page 5, the average lifetime of FAST-MAPbI₃ is two times longer than that of solution-synthesized single crystals, relevant test data should be provided, and the cited references should be updated (a 2016 reference cited here is too old). Likewise, in Figure 2B, the authors should provide relevant test data for single crystals.

Our response: We thank the reviewer for pointing out this question. We have updated the reference, added relevant data and discussions in the revised manuscript. Briefly, single crystal data of steady-state and transient PL are added in **Fig. 2b** and **2c**.

It should be noted that the presence of surface trap can significantly affect the photocarrier lifetime in the single-crystal samples. A recent paper¹¹ reveals <10 ns lifetime from the single crystal of MAPbI₃ synthesized from solution, which is smaller than prior reported value of 109 ns.

Adapted **Figs. 2b** and **2c** from the revised manuscript.

We also added relevant discussion in the main text (page 5).

“... Similar to the single crystal, the absorption edge of the FAST-MAPbI₃ reaches 852 nm, exceeding the edge of 780 nm of typical thin film counterparts, which can be ascribed to the indirect band transfer²⁵ that leads to smaller optical bandgaps in thicker samples. It should be noted that the FAST sample shows a strong anti-Stokes shift suggesting higher energy states existing in the PL process. The presence of these higher energy states are more likely due to the cubic phase and lattice strain induced Rashba splitting that enlarges the PL bandgap^{26,27} ...”

3. In line 155, page 5, the manuscript claims that the trap density and carrier mobility of FAST-MAPbI₃ are close to that of single crystals, but in fact the trap density is one order of magnitude lower than some single crystals, and the carrier mobility is two orders of magnitude lower (Fig. 2E). It is difficult to explain that the quality of FAST-MAPbI₃ crystal is comparable to single crystal. Please give a reasonable explanation.

Our response: We appreciate the reviewer's comment. To the best of our knowledge, prior studies have revealed a big difference in charge carrier mobility and trap density between polycrystalline film samples and single crystalline samples of MAPbI₃. Moreover, within each group of either polycrystal or single-crystal, there could be a large discrepancy in values because of the variation in sample quality especially the surface trap effect, chemicals used in the wet chemistry synthesis, and the measurement methods. For example, Liu et al¹² reported a trap density of 1.8×10^9 cm⁻³ and a carrier mobility of 34 cm²V⁻¹s⁻¹ by Hall effect measurements from an inverse temperature solution growth. While Dong et al¹³ reported a trap density of 3.6×10^{10} cm⁻³ and a carrier mobility of 164 cm²V⁻¹s⁻¹ by space-charge-limited-current from a top-seeded solution growth method.

In our manuscript (**fig. 2e**, adapted below), according to literature, the mobility can be 2 orders of magnitude difference in single-crystalline samples, and 5 orders of magnitude difference in polycrystalline samples. Therefore, we use a region of data sets from various typical works to make a more general comparison between polycrystals and single-crystals.

As for the FAST samples, due to its polycrystalline nature, at the beginning we anticipated its mobility and trap density similar to those prior reported polycrystalline samples. While after testing, the trap density is 4-order lower than that from solution-processed polycrystalline samples and mobility is exceeding $1 \text{ cm}^2\text{V}^{-1}\text{s}^{-1}$, close to that of single-crystalline samples. From the **fig. 2e** (cited below), we would conclude that the FAST samples having the properties that are closer to those of single crystals rather than polycrystals.

We appreciate reviewer’s concern on this, in the main text, we have revised certain descriptions to make it more scientifically strict.

For example, we have rephrased the following text.

“...When comparing to the data set from prior literature (trap density-mobility plot in **Fig. 2e**), we find these values of FAST-MAPbI₃ located in a region closer to the single crystals rather than typical polycrystalline films...”

Adapted **Fig. 2e** from the revised manuscript.

4. In line 180, page 6, the FAST MIM device has a PCE of 1.2%, only 60% of the single-crystalline lateral device (2% PCE), and it's hard to say that the two values are comparable.

Our response: We thank the reviewer for raising this question. First of all, we have revised the description in our manuscript on this valuable concern.

On page 7, we have revised the description as follows:

“...Albeit of the relatively low PCE, the J_{SC} and V_{OC} values of the FAST MIM device still approaches the values of the prior reported single-crystalline lateral device that has been optimized with surface trap passivation and charge selective buffer layers (J_{SC} and V_{OC} values of 0.78 V and 8.8 mA cm^{-2} , respectively)³⁶...”

It should be noted that our device utilized a simplest MIM structure without any structural or material optimization. This is simply for demonstration and proof-of-concept. While the prior paper utilized an optimized device structure with both surface trap passivation and buffer layer incorporation to modify the interface as well as to induce structural asymmetry for construction of an internal field. These structural optimizations can lead to the efficiency improvement. In our case, we directly deposit electrode on the FAST sample surface without optimization. This primitive device still shows PV parameters of V_{OC} of 0.71 V and

J_{sc} of 6.3 mA cm⁻² (approach to the 0.78 V and 8.8 mA cm⁻² from literature). We anticipate this FAST MIM holds the potential to higher efficiencies after proper optimization.

5. (1)The optical band gap of FAST-MAPbI3 is smaller than that of single crystals (line 141, page 5).

(2)The average lifetime of FAST-MAPbI3 is longer than that of single crystals (line 151, page 5).

(3)The Seebeck coefficient of FAST-MAPbI3 is higher than that of thin films and single crystals (line 196, page 7).

Please give reasonable explanations for the above three points.

Our response: Thank you for this comment. The single crystals synthesized from wet chemistry methods can have a few features: (i) solvent remaining within the crystal¹⁴, (ii) surface trap issue¹⁵, and (iii) point defect¹⁶ within the lattice due to incomplete assembly in the dynamic process within crystallization.

These effects can jointly lead to trap states near the band edge, disordering of the lattice that distorts the reciprocal lattice and consequently changes the electronic band structure, fluctuation of Fermi level (by point defect induced self-doping and /or impurity-doping effect). The band structure is corresponding to a few properties such as charge carrier mobility, effective mass of electrons and holes, bandgap, as well as phonon related behaviors. Therefore, these effects in real sample will deviate the optical bandgap, trap the carrier, and affect transport of carrier and even the thermodynamic of hot carriers. The difference from optical bandgap, average lifetime, and Seebeck coefficient can be joint results of the abovementioned effects, which is originated from the crystalline feature of the sample. The FAST-samples show very condensed feature, as indicated from both XRD and SEM results. Particularly, the remaining cubic phase as well as the lattice strain effect can be the reasons of these superior properties, compared with the case of typical solution-prepared tetragonal single crystals.

The main difference between single crystal and polycrystals is the grain boundary (GB). In principle, single crystal is free of GB, while polycrystals with GB may compromise optoelectrical properties. Because GB in typical inorganic semiconductors can scatter the charge carriers during their transport. This applies for most III-IV semiconductors. While for perovskite, the GB effect may not be as serious as those cases. Reports on polaron effect has provided a possible explanation of the defect-tolerated transport in these perovskites. The soft lattice and correlated electron-phonon coupling could provide a 'cage' to protect the charge carriers during the transport, i.e., the formation of a polaron. This can help carriers to transport cross the GB in perovskite. In fact, unlike Si-PV, the polycrystalline perovskite solar cell has higher efficiency than its single crystalline counterpart. From the perspective of the GB effect in perovskite materials, although there do exist GB in the FAST sample, the lower trap density in the sample can be the reason of these superior properties.

For example, we have added the following explanation on the single crystal-like feature.

On page 6

“...In general, the main difference between single crystal and polycrystal is the presence of grain boundary (GB) and lattice mismatch between neighboring grains. On the perspective of charge transfer, halide perovskites have been reported to have a polaron characteristic³¹ that can help charge transfer across long-range of disordered states such as GB, which mitigates the detrimental role of GB on transport. In parallel, solution processed single crystals have features such as the remaining solvent molecules within the crystal³², the presence of surface trap³³ and point defects³⁴. These features can jointly lead to additional trap states near the band edge, disordering of the reciprocal lattice that can change the electronic band structure and consequently the electrical properties (e.g., charge carrier mobility, effective mass). These factors from the downside of solution-grown single crystal and the upside of the FAST-polycrystal can lead to the comparable performance in Fig. 2e...”

6. The manuscript uses a lot of space (page 11, line 333 to page 12, line 357) to describe how to regulate the ratio of CsI and SnI₄, which is not related to the topic of the article, and the author is advised to move it into **Supplementary Materials**.

Our response: Thank you very much for this advice. We have moved this part into revised **Supplementary Information**, to **Supplementary Fig. 21**.

Relevant description has been changed in the main-text (page 12):

“...Interestingly, by using a stoichiometric ratio of precursor powder of CsI and SnI₄, we observed an impurity XRD peak of CsI at 2θ of 27.58° (**Fig. 4b(iii)**). To avoid so, we adjust the ratio of precursors and found that addition of extra 2.5 mol% of SnI₄ could stabilize the crystal. A more detailed discussion has been incorporated in **Supplementary Fig. 21**....”

Reviewer #3 (Remarks to the Author):

The manuscript by Zheng et al. reported the universal synthesis of halide perovskites via a field-assisted sintering technique that directly densifies the perovskites from solid precursors into high-quality bulk crystal within a few minutes. This method could be a powerful tool to prepare various halide perovskites. Although the manuscript showed many good results, some details, especially the mechanisms, should be further discussed and clarified. In this regard, a major revision is needed before being considered to get published in Nature Communications.

Our response: Thank you very much for spending time reviewing this work and we are also very grateful for the great comments and suggestions. All of these are greatly valuable and constructive in improving our manuscript. In the meantime, we also feel excited that the reviewer finds “This method could be a powerful tool to prepare various halide perovskites”. We understand reviewer #3 had a few concerns on the mechanism described in the manuscript, we have carefully added detailed explanation and discussion in the revised version. Please see the following details.

1. Why does the semiconducting and soft lattice nature of halide perovskites provide convenience for both gentle electrical field and mechanical field to perform the synthetic densification of perovskite simultaneously? The author should explain the mechanism clearly. Can this method be extended to other systems? Is it only suitable for materials with semiconductor and soft lattice properties?

Our response: We appreciate this comment and question.

The synthesis mainly takes advantage of (i) mechanical stress and (ii) internal current effect (localized joule heat) along the boundary of the powders. The mechanical stress can compress these powders into intimate aggregations, meanwhile the current can generate joule heating at the neck and intensified contact between neighboring powders.

In the case of halide perovskite materials, (i) the lattice of the halide perovskite is relative softer compared to those of metals and ceramics. This will allow sufficient deformation and densification of the perovskite under mild pressure (not very high pressure). (ii) And in the meantime, the semiconductor nature of halide perovskite such as MAPbI_3 , allows sufficient conductance of the current flowing internally through the whole green body of the sample. (iii) These two factors can also couple with each other during the synthesis, inducing secondary effects such as plasma sates to merge the boundary, facilitating the grain growth by the matter transfer along the “molten” or “fragmentized” boundaries and providing sufficient localized internal energy to re-assembly the crystal in a more thermodynamically favorable way. As a result, we found the resultant perovskite shows very low trap density with a magnitude of 10^{10} cm^{-3} and great grain size of $50 \mu\text{m}$ in the exemplified case of MAPbI_3 .

Based on these material points of the halide perovskite, we designed our facility to apply both mechanical stress and electrical current to the synthesis.

We appreciated reviewer for pointing this concern out, and we have added more descriptions in the revised manuscript (Page 3) to clarify the motivation of using both fields for the synthesis.

“The soft lattice nature allows sufficient deformation and densification of the perovskite powder precursors under mild pressure and the semiconducting feature provides sufficient conductance of the current flowing internally through the whole green body to trigger other secondary or coupling effects microscopically that can be uniformly distributed within the sample. For example, applying internal electric field that can induce joule heat homogeneously distributed throughout the sample and simultaneously applying a uniform pressure to condensate the perovskite, could be a potential way to obtain high-quality perovskite crystals.”

Application to other systems:

We believe this method can be potentially extended to other systems. While the principles here are (i) the presence of internal electrical current (localized joule heat along the boundaries and other secondary effects

such as plasma states), and (ii) certain degree of softness that can lead to strong densification (in fact, there could also be the concept of lattice strain effect due to the mechanical stress on these soft perovskite lattices (as pointed out by reviewer#1, we appreciated).

Following these two principles, other applicable system could be, for example, **semiconducting polymers that is of even softness features, other small molecular conjugated organics, metal-organic frameworks (MOFs) materials, and even new materials with both (semi-)conductive and soft natures**. In the extreme cases, for example, for the type of high-conductive and soft materials, the applied electrical current and mechanical force could be controlled at low value condition (saving energy for manufacturing). Nevertheless, it should also be considered about the coupling effect between the FAST processing parameters (current, pressure, time) and the material's intrinsic physical properties such as glass transition, melting, and decomposition temperatures, heat capacity, heat transfer properties, etc. There could be many trial and errors to tune the FAST processing parameters for each material system according to what we have observed in our halide perovskite system. On the other hand, when it comes to **low-conductive and rigid materials, such as oxide perovskite ceramics**, it would need a high temperature and long annealing time (several hours, according to literature of cold-sintering-post-annealing process of ceramics¹⁷). Generally speaking, it would be doable for broader systems, but quantitatively it may need extreme conditions such as ultrahigh temperature and huge mechanical stress. In anyway, from this work, our observation is that halide perovskite material class is a good target using this method.

2. As mentioned in the manuscript, “This surface heating effect, plus the low ionic activation energy (e.g., 0.1-0.6 eV of MAPbI₃) in halide perovskites, triggers the mass transfer and grain merging between neighboring powders”, “The uniaxial mechanical stress further enhances these effects”. According to these statements, the electric field is a unidirectional electric field and uniaxial mechanical stress, I wonder whether the perovskite is subjected to uniform stress in the whole system. If there is only uniaxial stress, the as-prepared perovskite should exhibit gradient density. Therefore, this approach might face some problems, such as whether the crystal quality and grain boundary of the upper and lower perovskites are different, which will seriously affect the properties and device performance.

Our response: We greatly appreciate the reviewer's comment. We are sorry for this ambiguous message in our original manuscript. Same to the electrical field, the mechanical stress executed on the sample is also “unidirectional”. Our original expression of “uniaxial” is to denote the stress direction from the plunger head. That is, the driving stress from the plunger is vertically applied to the sample for the setup convenience in our lab (this is schematized in the following **Fig. R6** below, also revised in **Fig. 1b** in the revised manuscript). We also changed the expression of “uniaxial” to “unidirectional” for the driving force from the plunger in the revised manuscript.

On page 4, the relevant content has been changed as follows:

“...The unidirectional mechanical stress that is statically and uniformly applied to the plunger head further enhances these effects...”

Force uniformity:

(i) Along x-y-plane: as the plunger covering all the top and down surface area of the sample, the pressure is executed uniformly to the sample. Therefore, along the x-y-plane (in the plane parallel to the sample surface, i.e., x-y-plane in the following **Fig. R6**), the sample is under uniform stress. **(ii) Along z-direction:** because the forces were slowly applied, the pressing process is in a static loading condition (the force is in the condition of equilibrium along the z-direction as well). Therefore, the powders are being executed with the force of the identical equilibrium condition in all the x-, y-, z-directions. From these aspects, we believe the sample has a uniform sintering condition.

Additional SEM results: To verify this, we carried out a SEM cross sectional characterization on a sample. Nine regions spreading over the cross-section have been taken from the sample. **Fig. R7a** (below) shows the SEM image of a cross-section of a FAST-MAPbI₃ sample. We take 9 regions marked by Roman numerals and the corresponding magnified images are shown in **Fig. R7b**. All the results show identical

crystalline feature (grain size, dense packing, no specific orientation or visible pores, etc.). This also applies to our observations on other samples in this study. Hence, we believe this method would not cause uniformity issues.

Fig. R6 Processing diagram showing the solid densification of halide perovskites in the graphite die during sintering. The voltage and mechanical force are uniformly applied onto the graphite plungers. The final diameter of bulk sample is the same as the graphite die, while the thickness of bulk sample is smaller than the loaded powder after densifying.

Fig. R7 Uniformity validation of cross-sectional SEM images of a FAST-MAPbI₃ sample. **a** SEM image at low magnification showing the overall feature at the cross-section of the sample. **b** Magnified local SEM images at different locations marked by Roman numerals in accordance to **Fig. R7a**.

3. The data presented in the manuscript indicate that the as-prepared perovskite is polycrystalline. There is no relevant mechanism explanation for why the performance of the as-prepared device based on such polycrystalline perovskites in figure 2 is comparable to that of the single crystal. In addition, the authors should provide some TEM data to prove the crystallization quality.

Our response: We thank the reviewer for this comment. We have added certain explanation in the main text. And following content shows the detailed reasons as well as newly added TEM results.

Mechanisms explanation (why the performance of FAST sample device similar to those of single crystals) can be from two scenarios of (1) prior single crystal from solution and (2) FAST polycrystal.

(1) Solution grown single crystal: the vast majority of halide perovskite single crystals were grown from solution process, where there are a few concerns we think may hinder the electrical properties. (i) solvent molecule/impurity remaining within the lattice. Solvation of precursors into the solution suggests there are certain interactions between the precursors and the solvent molecules. This interaction can also lead to extra solvent molecules to be involved within the lattice of perovskite when the precursor atoms assembly into the crystal during the growth process (as schematized as follows in the assembly cartoon in **Fig. R8**). This solvent-precursor interaction will also cause the looseness of atomic assembly and further lead to (ii) surface traps/micro-structures (as shown in the following **Fig. R8**), and (iii) point defects (e.g., Schottky/Frenkel defects) within the single crystal. It has been reported¹⁵ that the surface trap can significantly affect the single crystalline device performance, as it could hamper the interfacial charge transfer when contacting with other materials. Therefore, although single crystal is free of grain boundary and maintain the same facet orientation of the lattice throughout the sample, there still remains these three factors that can compromise the overall property. This is also consistent to the observation that even in the same composition, single crystals from different solution methods can lead to different trap density and carrier mobility varying over 2 orders of magnitudes.

(2) FAST polycrystal: The major difference between polycrystal and single-crystal is the grain boundary (GB), which is a discontinuous gap between neighboring grains. A typical understanding of GB is an inhibitor for charge carrier transfer across it. This is valid in typical III-V semiconductors. However, halide perovskite has been reported to have certain “tolerance”^{18,19} on the disorder states such as the GB, and some researches also reveals the beneficial role²⁰ of the GB. This can be understood by a “polaron model” that the electron-phonon coupling enables certain protection of the active carrier (lattice deforms with the charge carrier by this electron-phonon coupling which provides a “cage” to protect the “free” electron) which allows it to hop over even a GB from grain to grain. A good fact to support the trivial detrimental role of GB can be seen from solar cell efficiency. Perovskite polycrystalline solar cell device can reach to PCE over 25% which is even higher than their single-crystalline counterpart. While for silicon, mono-Si holds the record of 26% that is higher than its polycrystalline counterpart (20%, and much higher than 11.9% from microcrystalline Si solar cell). Nevertheless, GB can be of many manifestations. That is, if we look into the quality of GB, it could be of different types such as a discontinuous gap, an amorphous phase consisting of ions, impurities, and vacancies, a disordered phase of large lattice distortion/strain/incompletion, or simply an interface of two grains of different orientations, or any combination of these. The more compact the grains are packing, the easier for the charge carriers to transfer across the GB. This can also apply to our FAST samples. We have shown in the SEM images, 10-100 μm scale grains are presented in FAST-MAPbI₃ with closely packed feature. How closely the grains are packed will affect how efficient the charge transfer. The FAST provides the possibility to make the grain solidified and merged in high degrees, as we mentioned above on the mechanical and electrical field application in the synthesis. Thus, we would ascribed comparable performance to both the highly compact grain and low defect concentration in our FAST sample as well as the trivial detrimental role of highly squeezed GB by the FAST.

Fig. R8 Multiple micro-structures, surface traps, and hypothetical vacancy cites during the single-crystal growth from solution. Adapted with permission from © The Royal Society of Chemistry 2015, © The Royal Society of Chemistry 2018.

We added related discussion in the revised manuscript as follows (Page 6 & 7):

“In general, the main difference between single crystal and polycrystal is the presence of grain boundary (GB) and lattice mismatch between neighboring grains. On the perspective of charge transfer, halide perovskites have been reported to have a polaron characteristic³¹ that can help charge transfer across long-range of disordered states such as GB, which mitigates the detrimental role of GB on transport. In parallel, solution processed single crystals have features such as the remaining solvent molecules within the crystal³², the presence of surface trap³³ and point defects³⁴. These features can jointly lead to additional trap states near the band edge, disordering of the reciprocal lattice that can change the electronic band structure and consequently the electrical properties (e.g., charge carrier mobility, effective mass). These factors from the downside of solution-grown single crystal and the upside of the FAST-polycrystal can lead to the comparable performance in **Fig. 2e**.”

Thanks for the suggestion on using TEM to verify the crystallographic information of the sample. Accordingly, we have further conducted the TEM measurement and added the related results in **Supplementary Fig. 5**.

For your convenience, the related contents are also shown here:

“Adapted **Supplementary Fig. 5i-k** from the revised **Supplementary Information**.”

Related caption and description:

“(i-k) TEM results on FAST-MAPbI₃ sample: i schematic and atomic arrangement of a pseudo-cubic MAPbI₃ lattice. Heavy atoms of I and Pb are noted in the figure with a d-spacing of 0.28 nm

of (310) and an interplanar angle of 120° . **j** an TEM image of the FAST-MAPbI₃ sample with the same perspective to that in **(i)**, showing identical values of d-spacing and interplanar angle to the molecular model in **(i)**. **k** TEM image of other locations in the same sample. Inset: the corresponding fast Fourier transform (FFT) image. We take two positions of “1” (blue) and “2” (red), and obtain the d-spacing of (040) and ($\bar{2}$ 02) planes with values of 0.22 and 0.36 nm, respectively. This is in consistent to the crystal feature of the MAPbI₃.”

4. I do not quite agree with the relevant description about "heterojunction", which is more like a compound than a heterojunction. As shown in S17, at the interface of the MAPbCl₃ and FASnI₃, the density of the two materials is obviously different, and there are a lot of voids. The as-prepared material appears to show poor compactness. In addition, the synthesized heterojunction is too thick, can the perovskite heterojunction with few layers and periodic stacking be prepared?

Our response: We appreciate the reviewer's comment that makes us re-consider the junction. To the best of our knowledge, the “heterojunction” is defined as “a junction between two different layers or regions of crystalline semiconductors have unequal band gaps.” Thanks to reviewer#3's comment, we re-checked the junction using cross-section SEM (these data have been added to **Supplementary Fig. 17a** in the revised **Supplementary Information**, also shown below for your reading convenience). There seem not much visible voids at this junction (the junction between two materials remains compact and densified). We agree with reviewer that the crystalline feature (e.g., grain size) deviates between MAPbCl₃ and FASnI₃. However, this different grain sizes of each layer would not affect the compactness. On the contrary, from the EDS results, we found along the vertical direction, all the elements exhibit gradient feature starting from the junction. For example, Sn element from the bottom FASnI₃ layer also show some presence in the upper layer (**Supplementary Fig. 17c**). This information indicates the junction is not simply a stacking of two components. Instead, the interpenetration feature can be an evidence of good heterojunction.

Overall, the main purpose of making this junction of MAPbCl₃/FASnI₃ is to show the manufacturing compatibility and potential application of this FAST method in making this junction as an inspiration for future use. To uniformize the crystal size, or making a smoother or even meso-/micro-structured hetero-interface, there could be further optimizations using various strategies such as using lattice-matching components, designing interfacial structure (patterned surface, etched surface like silicon pyramid etc.), optimizing experimental parameters (temperature, pressure, time, etc.).

Adapted Supplementary Fig. 17 MAPbCl₃/FASnI₃ heterojunction. a cross-sectional SEM images showing multiple sites of the hetero-interface. **b** Magnified cross-sectional SEM at the hetero-interface, showing an intimate interface between different perovskites. **c** EDS mapping of different elements, indicating two perovskites closely joining together at the interface.

Periodic stacking

The thickness of the junction layer is controlled by the amount of powder loaded into the die. Generally, we add ~1 g (depends on the density) halide perovskite powders into a die (diameter of 0.5 inch) to fabricate the sample with thickness ~2 mm. If the required thickness of periodic stacking layer is in millimeter-scale, it's doable by loading powders manually. However, to make thinner stacking layer thickness, there will need additional tools to pave the powder uniformly on top of each other.

We understand for certain applications such as PV, it would consider the thickness to be smaller than the diffusion length (typically less than hundreds of micrometers for perovskite). While this method is mainly for bulk sample synthesis. Slicing from top-down methods could be useful to modulate the thickness. But for specific applications such as piezoelectric transducer, thermoelectric, X-ray detection which requires large thickness bulk sample, this method could be more effective.

5. The universality section should provide more data to demonstrate the advantages of this method. More than one perovskite of each type should be prepared, and the current data are not sufficient to prove that this is a general method.

Our response: We appreciate reviewer for this comment. In our original manuscript, we have attempted different types of the halide perovskite related FAST-samples. During our experiment, each composition has been carefully adjusted with various processing parameters (time of sintering and time of cooling ramp, electrical current applied, loading conditions such as die shape, geometries, and post-loading compressing, etc.) to condense the sample into good crystalline feature.

According to the comment, we also added additional demonstrations on different compositions for each type. As can be seen in the revised **Supplementary Table 5** (also shown below for your reading convenience), totally over 14 major compositions (including more sub-group compositions such different stoichiometric ratios of precursors in synthesis of Cs₂SnI₆, the concentration modulation in perovskite-Bi₂Te₃ alloy, etc.) have been tested. Overall, we find all these compositions can lead to great crystalline feature through this FAST method (100% success rate for over 50 individual trials).

The revised **Supplementary Table 5** is shown here for reading convenience.

Supplementary Table 5 Summary of FAST-synthesized perovskites in this work.

Materials	Composition	Precursors	FAST	Product	Density
Prototype	MAPbI ₃	MAI & PbI ₂	52 MPa; 200 °C		4.16 g cm ⁻³
Alloy	(Bi ₂ Te ₃) _{0.1} (MAPbI ₃) _{0.9}	MAPbI ₃ & Bi ₂ Te ₃	52 MPa; 250 °C		4.24 g cm ⁻³
	(Bi ₂ Te ₃) _{0.9} (MAPbI ₃) _{0.1}				6.36 g cm ⁻³
heterojunction	MAPbI ₃ /conductive polymers	MAPbI ₃ & conductive compound (ProbeMet™)	29 MPa; 150 °C		/
	MAPbCl ₃ /FASnI ₃	MACl & PbCl ₂ /FAI & SnI ₂	52 MPa; 200 °C		/
Lead-free	MASnI ₃	MAI & SnI ₂	52 MPa; 150 °C		/
	CsSnI ₃	CsI & SnI ₂	52 MPa; 300 °C		4.59 g cm ⁻³
	FASnI ₃	FAI & SnI ₂	52 MPa; 150 °C		/

2D	$(MA)_3Bi_2I_9$	MAI & BiI ₃	52 MPa; 300 °C		4.10 g cm ⁻³
	$MA_2CuCl_2Br_2$	MACl & CuBr ₂	52 MPa; 120 °C		/
All-inorganic	$Cs_3Bi_2I_9$	CsI & BiI ₃	52 MPa; 500 °C		4.62 g cm ⁻³
	Cs_2SnI_6	CsI & SnI ₄	52 MPa; 300 °C		4.73 g cm ⁻³
Color (larger bandgap)	MAPbCl ₃	MACl & PbCl ₂	52 MPa; 200 °C		/
	FAPbBr ₃	FABr & PbBr ₂	52 MPa; 180 °C		/

6. The experimental content related to the stability of perovskite prepared by this method should be added to the manuscript.

Our response: We thank reviewer#3 for this stability question.

We have added the stability test on the prototype MAPbI₃, and the easily oxidized FASnI₃ samples for the stability verification. This information has been added in this revised version. Details are shown below:

MAPbI₃: Even after storing in the ambient for 2 months (samples we prepared in July), the XRD results still exhibit identical spectrum to that of the fresh sample (no degraded impurities such as PbI₂ is detected). By magnifying the 2theta range accounting for tetragonal (004) and (220) peaks, we observed that the original cubic phase gradually converts into the tetragonal phase. This can be further visualized in **Supplementary Fig. 25c**, that the (200) cubic peak in the fresh sample has an identical peak intensity to the neighboring (220) tetragonal peak. While after 2 months, the (220) tetragonal peak shows relatively stronger intensity than the (200) cubic peak.

FASnI₃: It is shown that in the case of storage inside of a glovebox, even after 1 year, the sample does not show any impurities or degradation peaks in the XRD. While storing in the ambient air, after 12 h there is emergence of SnI₄ phase, and after 48 h there is emergence of SnO₂ phase. This evolution is consistent to the sequential degradation process in the equation set in **Supplementary Fig. 26b**. While from the photo, it is hard to observe obvious color change of the sample. This observation, together with the remaining scattering peaks of FASnI₃ in the 48-h age sample in **Supplementary Fig. 26a**, suggest there could be a protection layer formed by the degrades covering the surface, which can prevent further degradation of the inner sample.

Supplementary Fig. 25 Stability study of the FAST MAPbI₃ samples. **a** XRD results of a EM-FAST MAPbI₃ pellet stored in the ambient atmosphere for different time (RH 35-80%, depending on daily whether at State College, PA, United State, temperature of ca. 25 °C controlled by lab). **b** Local magnification of the XRD results, showing the phase evolution from cubic (200) planes towards tetragonal (004) and (220) planes. **c** Comparison of fresh and 2-month age XRD, displaying the phase evolution. These results suggest robust feature of the EM-FAST MAPbI₃ samples (no obvious degradation even after ambient storage for 2 months). The origin may come from the hypothetical lattice strain in these samples, which could increase the activation energy of ion migration^{32,33}.

Supplementary Fig. 26 Stability study of the FAST FASnI₃ samples. **a** XRD results of a EM-FAST FASnI₃ pellet stored in the ambient atmosphere for different time (RH 35-80%, depending on daily whether at State College, PA, United State, temperature of ca. 25 °C controlled by the lab). **b** Photography of the FASnI₃ sample with different ages from 0 h to 24 h, exposed in ambient air. A degradation chemical reaction set is inserted to show the compositional evolution.

First of all, it is shown that in the case of storage inside of a glovebox, even after 1 year, the sample does not show any impurities or degradation peaks in the XRD. While storing in the ambient air, after 12 h there is emergence of SnI₄ phase, and after 48 h there is emergence of SnO₂ phase. This evolution is consistent to the sequential degradation process in the equation set in **Supplementary Fig. 26b**. While from the photo, it is hard to observe obvious color change of the sample. This observation, together with the remaining scattering peaks of FASnI₃ in the 48-h age sample in **Supplementary Fig. 26a**, suggest there could be a protection layer formed by the degrades covering the surface, which can prevent further degradation of the inner sample.

7. The TRPL exhibits two decay paths, the slow and fast components should be fitted separately, and an explanation should be provided.

Our response: We thank reviewer#3 for this comment.

As can be seen in **Supplementary Note 1 TRPL average lifetime**. We have incorporated physical explanation of this information.

Fitting Algorithm

For the TRPL fitting process, we utilized the bi-exponential decay function of

$$f(t) = A_1 \exp\left(\frac{-t}{\tau_1}\right) + A_2 \exp\left(\frac{-t}{\tau_2}\right) + B \quad (\text{S2})$$

where τ_1 is the fast decay lifetime component, A_1 is the fast decay amplitude, τ_2 is the slow decay lifetime component, A_2 is the slow decay amplitude, and B is a constant, respectively. Levenberg–Marquardt algorithm (LMA) is used for seeking the best fitting parameters, and evaluated by the parameter of:

$$\chi^2 = \sum_k w_k^2 \frac{[f_k - F_k]^2}{n} \quad (\text{S3})$$

Where the w_k is the weighting factor, f_k is the fitting value and F_k is the experimental value, n is the number of free parameters approximately to the number subtracted from the fitted data points by the number of lifetime parameters used in the fitting. χ^2 has a theoretical limit 1.0 for Poissonian distributed and in typical, χ^2 needs to be less than 1.0 to secure a good fitting. Higher order exponential decay can lead to a smaller χ^2 , i.e., better fitting. While choosing the decay model also needs to take consideration of the sample condition. In this work, we utilized the bi-exponential decay because it already exhibited a good fit by showing χ^2 of < 0.5 . Additionally, for MAPbI₃, it has been widely reported both surface trap and bulk defect can lead to relative quick and slow decay component in the TRPL²¹. This well-agreed model is also consistent to the result observed in this work. Hence, we employ the bi-exponential decay function to quantify these underlying attributes of the samples of interest.

The following contents have been added in the revised version:

“Briefly, for halide perovskites, both surface trap (faster component) and bulk defect (slower component) could contribute different population to the overall fluorescence. As can be seen in the **Supplementary Table 1**, all the three samples of thin-film, powder, and FAST-MAPbI₃ show similar ratio of A1 and A2, which makes it easier to directly compare faster and slower lifetime. Along the surface trap-induced decay, the FAST-samples display the longest lifetime of 72.4 ns compared to thin film (30.2 ns) and powder (6.7 ns). This is consistent to the high trap density nature of the powder and indicates the FAST sample has less surface trap detrimental effect compared to the film sample. Similarly, from the perspective of bulk trap induced decay, the FAST sample also shows longest lifetime of 311 ns, compared to 63 and 77 ns of powder and thin film samples, respectively. This also suggests the high quality of the FAST sample in bulk. While for single crystal sample, as the ratio of A1 and A2 is different from the other three samples, it is hard to directly compare the τ_1 and τ_2 . Hence, the τ_{ave} can be a more proper figure-of-merit to evaluate the lifetime. Apparently, single crystal sample shows a smallest value of 20.7 ns. This is more likely due to the presence of hypothetical solvent impurity, surface microstructure and point defect as discussed in previous text in this response file.”

Supplementary Table 1 Parameters of TRPL measurement for thin film, powder, and FAST-MAPbI₃.

Sample	A ₁	τ_1 (ns)	A ₂	τ_2 (ns)	τ_{ave} (ns)
--------	----------------	---------------	----------------	---------------	-------------------

Thin film-MAPbI ₃	0.57	30.2	0.43	77	61.1
Powder-MAPbI ₃	0.63	6.7	0.37	63	54.3
Single crystal-MAPbI ₃	0.18	1.4	0.82	25	20.7
FAST-MAPbI ₃	0.56	72.4	0.44	311	257.1

Reference used in this response

1. Tie, S. *et al.* Robust Fabrication of Hybrid Lead-Free Perovskite Pellets for Stable X-ray Detectors with Low Detection Limit. *Adv. Mater.* **32**, 2001981 (2020).
2. Yang, B. *et al.* Heteroepitaxial passivation of Cs₂AgBiBr₆ wafers with suppressed ionic migration for X-ray imaging. *Nat. Commun.* 2019 **10**, 1–10 (2019).
3. Shrestha, S. *et al.* High-performance direct conversion X-ray detectors based on sintered hybrid lead triiodide perovskite wafers. *Nat. Photonics* 2017 **11**, 436–440 (2017).
4. Stoumpos, C. C., Malliakas, C. D. & Kanatzidis, M. G. Semiconducting Tin and Lead Iodide Perovskites with Organic Cations: Phase Transitions, High Mobilities, and Near-Infrared Photoluminescent Properties. *Inorg. Chem.* **52**, 9019–9038 (2013).
5. Bonadio, A. *et al.* Entropy-driven stabilization of the cubic phase of MAPbI₃ at room temperature. *J. Mater. Chem. A* **9**, 1089–1099 (2021).
6. Wang, K. *et al.* A Nonionic and Low-Entropic MA(MMA)_nPbI₃-Ink for Fast Crystallization of Perovskite Thin Films. *Joule* **4**, 615–630 (2020).
7. Liu, D. *et al.* Strain analysis and engineering in halide perovskite photovoltaics. *Nat. Mater.* 2021 **20**, 1337–1346 (2021).
8. Xue, D. J. *et al.* Regulating strain in perovskite thin films through charge-transport layers. *Nat. Commun.* 2020 **11**, 1–8 (2020).
9. Wang, T. *et al.* Indirect to direct bandgap transition in methylammonium lead halide perovskite. *Energy Environ. Sci.* **10**, 509–515 (2017).
10. Frohna, K. *et al.* Inversion symmetry and bulk Rashba effect in methylammonium lead iodide perovskite single crystals. *Nat. Commun.* 2018 **9**, 1–9 (2018).
11. Wang, W. *et al.* Defect Healing of MAPbI₃ Perovskite Single Crystal Surface by Benzylamine. *Symmetry* 2022, Vol. 14, Page 1099 **14**, 1099 (2022).
12. Liu, Y. *et al.* Two-Inch-Sized Perovskite CH₃NH₃PbX₃ (X = Cl, Br, I) Crystals: Growth and Characterization. *Adv. Mater.* **27**, 5176–5183 (2015).
13. Dong, Q. *et al.* Electron-hole diffusion lengths > 175 μm in solution-grown CH₃NH₃PbI₃ single crystals. *Science* **347**, 967–970 (2015).
14. Zhou, Y. *et al.* Effect of Solvent Residue in the Thin-Film Fabrication on Perovskite Solar Cell Performance. *ACS Appl. Mater. Interfaces* **14**, 28729–28737 (2022).
15. Wang, X. D. *et al.* Surface passivated halide perovskite single-crystal for efficient photoelectrochemical synthesis of dimethoxydihydrofuran. *Nat. Commun.* 2021 **12**, 1–9 (2021).
16. Lei, Y., Chen, Y. & Xu, S. Single-crystal halide perovskites: Opportunities and challenges. *Matter* **4**, 2266–2308 (2021).
17. Li, M. *et al.* A family of oxide ion conductors based on the ferroelectric perovskite Na_{0.5}Bi_{0.5}TiO₃. *Nat. Mater.* 2013 **13**, 31–35 (2013).
18. Kim, G. W. & Petrozza, A. Defect Tolerance and Intolerance in Metal-Halide Perovskites. *Adv. Energy Mater.* **10**, 2001959 (2020).
19. Das, B., Liu, Z., Aguilera, I., Rau, U. & Kirchartz, T. Defect tolerant device geometries for lead-halide perovskites. *Mater. Adv.* **2**, 3655–3670 (2021).
20. P Adhyaksa, G. W. *et al.* Understanding Detrimental and Beneficial Grain Boundary Effects in Halide Perovskites. *Adv. Mater.* **30**, 1804792 (2018).
21. Hou, Y. *et al.* Enhanced Performance and Stability in DNA-Perovskite Heterostructure-Based Solar Cells. *ACS Energy Lett.* **4**, 2646–2655 (2019).

REVIEWERS' COMMENTS

Reviewer #1 (Remarks to the Author):

The authors have made significant revisions to the MS, and have addressed my comments adequately. It is now in a great shape for publication, I thus support its publication in nature comm without further changes.

Reviewer #2 (Remarks to the Author):

The authors answered our questions, and the revised manuscript is more suitable for publication in Nature Communications. Some references need to be updated before the manuscript is published.

On line 339, page 11, the authors constructed a heterojunction-structured perovskite (MAPbCl₃/FASnI₃) by FAST. The authors are suggested to read and consider this paper (DOI: 10.1016/j.joule.2022.04.012) which studied the heterojunction structure of perovskites.

Reviewer #3 (Remarks to the Author):

The authors have revised this manuscript carefully and responded the reviewers' comments well. The revised manuscript can be accepted for publication.

Reviewers' comments are highlighted in brown.

Our responses are in black.

The additional or revised sentences cited from the revised manuscript or Supplementary Information are highlighted in blue.

REVIEWER COMMENTS

Reviewer #1 (Remarks to the Author):

The authors have made significant revisions to the MS, and have addressed my comments adequately. It is now in a great shape for publication, I thus support its publication in nature comm without further changes.

Our response: We greatly appreciate reviewer #1 reviewing our response and revised manuscript. Thank you very much for the comments.

Reviewer #2 (Remarks to the Author):

The authors answered our questions, and the revised manuscript is more suitable for publication in Nature Communications. Some references need to be updated before the manuscript is published. On line 339, page 11, the authors constructed a heterojunction-structured perovskite (MAPbCl₃/FASnI₃) by FAST. The authors are suggested to read and consider this paper (DOI: 10.1016/j.joule.2022.04.012) which studied the heterojunction structure of perovskites.

Our response: Thanks for providing us this valuable information. We have added this reference on page 11 as ref 47:

"In addition, we also constructed a heterojunction structured perovskite by FAST. It should be noticed that there is rare report on the ultrathick hetero-bilayer of halide perovskites.⁴⁷"

Reviewer #3 (Remarks to the Author):

The authors have revised this manuscript carefully and responded the reviewers' comments well. The revised manuscript can be accepted for publication.

Our response: Thank you very much for reviewing our revision as well as providing the comment.